# Structural basis for activity switching in polymerases determining the fate of let-7 pre-miRNAs

Gangshun Yi [1,2], Mingda Ye [3], Loic Carrique [1], Afaf El-Sagheer [4,8], Tom Brown [4], Chris J. Norbury [5], Peijun Zhang [1,6,7] & Robert J. C. Gilbert [1,2] ✉

Tumor-suppressor let-7 pre-microRNAs (miRNAs) are regulated by terminal uridylyltransferases TUT7 and TUT4 that either promote let-7 maturation by adding a single uridine nucleotide to the pre-miRNA 3′ end or mark them for degradation by the addition of multiple uridines. Oligo-uridylation is increased in cells by enhanced TUT7/4 expression and especially by the RNA-binding pluripotency factor LIN28A. Using cryogenic electron microscopy, we captured high-resolution structures of active forms of TUT7 alone, of TUT7 plus pre-miRNA and of both TUT7 and TUT4 bound with pre-miRNA and LIN28A. Our structures reveal that pre-miRNAs engage the enzymes in fundamentally different ways depending on the presence of LIN28A, which clamps them onto the TUTs to enable processive 3′ oligo-uridylation. This study reveals the molecular basis for mono- versus oligo-uridylation by TUT7/4, as determined by the presence of LIN28A, and thus their mechanism of action in the regulation of cell fate and in cancer.

Eukaryotic RNAs are subject to chemical modification and editing from the beginning of transcription onward[1,2]. While some modifications such as the methylation of adenine and guanine residues are very well established[1,2], others including modification of the 3′ tails of RNA species are only more recently described[3,4]. In particular, the nontemplated addition of specific and different selections of nucleotides by a family of terminal nucleotidyltransferase enzymes has been found to have important roles in the maturation and degradation of messenger RNAs (mRNAs) and microRNAs (miRNAs)[4,5]. In turn, these modifications have been found to have critical input to mechanisms regulating inflammation, viral replication and disease, cell division and development, and cancer[6–13]. A well-defined example involves the 3′ uridylation of pre-let-7 miRNAs. During miRNA expression, a primary (pri-miRNA) form is first processed into a pre-miRNA within the nucleus, before the cytoplasmic endoribonuclease Dicer matures the miRNA: one strand of which is loaded onto an Argonaute protein to generate an RNA-induced silencing complex[5,14]. Pre-miRNAs consist of a 72–75 nucleotide (nt) hairpin loop with a 3′ overhang, which for efficient processing is 2 nt longer than the 5′ end, as detected by Dicer[5,14]. Ninety percent of pre-let-7 miRNAs are of a Class II group that has only a single-nucleotide 3′ overhang, and therefore require addition of a single uridine-monophosphate moiety for efficient onward processing[15–17]. By contrast, oligo-uridylation results in a pre-miRNA that Dicer cannot process[5,15–17], and which is subsequently degraded by the exonuclease Dis3l2 (refs. 18,19).

In *Homo sapiens*, two homologous enzymes terminal uridylyltransferase 4 (TUT4; ZCCHC11) and TUT7 (ZCCHC6) are involved in both mono- and oligo-uridylation of (let-7) pre-miRNAs as well as of histone mRNAs and some other targeted mRNAs[3,4,16,17,20,21]. Alone, TUT4 and TUT7

[1]Division of Structural Biology, Centre for Human Genetics, Nuffield Department of Medicine, University of Oxford, Oxford, UK. [2]Calleva Centre for Evolution and Human Science, Magdalen College, Oxford, UK. [3]Centre for Medicines Discovery, University of Oxford, Oxford, UK. [4]Chemistry Research Laboratory, University of Oxford, Oxford, UK. [5]Sir William Dunn School of Pathology, University of Oxford, Oxford, UK. [6]Diamond Light Source, Harwell Science and Innovation Campus, Didcot, UK. [7]Chinese Academy of Medical Sciences Oxford Institute, University of Oxford, Oxford, UK. [8]Present address: Institute for Life Sciences, University of Southampton Highfield Campus, Southampton, UK. ✉e-mail: robert.gilbert@magd.ox.ac.uk

tend to engage in a distributive activity yielding mono-uridylation[16,17,20], which has also been characterized in their yeast homolog Cid1 (ref. 22). However, in the presence of the RNA-binding factor LIN28A, TUT4/7 gain a processive activity, leading to oligo-uridylation[16,17]. The importance of the let-7 miRNA+TUT4/7-LIN28A axis to cell biology and cell fate is highlighted by the large body of evidence linking it in particular to a variety of cancers[17,23–26], but also to metabolic regulation[27] and inflammatory responses[13]. Let-7 miRNAs exert their anti-proliferative and tumor-suppressor effects by targeting the mRNAs of a host of oncogenic proteins, including MYC, the RAS family, BCL-2 and LIN28A[5] (MYC also downregulates let-7 signaling, via an mRNA sponging mechanism[28]). Clinical data indicate a correlation between reduced let-7c and poor prognosis in non-small cell lung carcinoma[29]; and TUT4 over-expression is associated with reduced overall survival in primary breast cancer[30]. TUT4 short-hairpin RNA knockdown in human hepatoma, melanoma, breast, ovarian and prostate cancer cells inhibits growth and invasiveness in culture and in xenografts, as does direct reintroduction of let-7 miRNAs[17,31]. The effects of TUT4 and TUT7 are proving to be context and (cancer) cell type dependent[21,32].

Here we describe structures of full-length human TUT7 and near-full-length TUT4, as determined by cryogenic electron microscopy (cryo-EM) and single-particle analysis, at resolutions of up to 3.53 Å. Four structures are of TUT7; alone (apo), bound to pre-let-7g miRNA (binary complex) and in two states with both the pre-let-7g miRNA and LIN28A. The fifth structure is of human TUT4 bound with pre-let-7g and LIN28A. Together, the structures we describe reveal the basis for mono- or oligo-uridylation via alternative enzymatic conformations and protein−RNA interfaces, induced either by the selective binding of a pre-let-7 miRNA, or by the initial binding of the pre-let-7 miRNA to LIN28A, followed by engagement of both the pre-miRNA and LIN28A itself directly with TUT7/4. As such, they demonstrate the molecular basis of LIN28A's function in driving cellular transformation.

## Results and discussion
### Cryo-EM structure determination of apo TUT7 and TUT7/4 complexes

TUT4 and TUT7 are multidomain enzymes whose specific targeting of RNAs makes use of three zinc fingers, combined with two copies of a nucleotidyltransferase domain (NTD), one inactive in the N-terminal half of the polypeptide (pseudo-NTD) and one active in the C-terminal half (Fig. 1a,b)[4,5]. Structures have previously been determined for the N-terminal LIN28A-interacting module (LIM) of TUT4 including its pseudo-NTD domain[33], and for the C-terminal catalytic module (CM) of TUT7 (ref. 34), as well as for LIN28A bound with RNA constructs representing pre-let-7d/f/g miRNAs[35,36]. The human TUT4 LIM[33] was found to have a very similar basic architecture to that of the TUT7 CM[34]. The TUT4 LIM structure was, however, distinguished by the presence of its zinc finger domain at the N terminus, which is homologous to those of double-stranded RNA-binding zinc fingers[33]. The TUT7 CM structure was previously solved crystallographically in apo and complex forms with double-stranded RNA, $U^2$, $U^5$ or uridine triphosphate (UTP)[34]. In these CM structures, the second TUT zinc knuckle (ZK2) could be visualized bound with both $U^2$ and $U^5$. The diverse substrate complexes revealed that the RNA-binding pocket orients a group II pre-let-7 to favor mono-uridylation, while the ZK2 of TUT7 is thought to aid oligo-uridylation by supporting the extension of the oligo-U tail[34]. However, none of the structures determined so far have been able to show the mechanistic basis for the alternative activities shown by TUT7 and TUT4: that is, mono- versus oligo-uridylation.

LIN28, as a key regulator controlling the timing of developmental events and participating in heterochronic processes, is highly conserved across diverse organisms[37]. In *C. elegans*, there is only one encoded LIN28 protein, while in humans there are two paralogs, LIN28A and LIN28B[17,37]. LIN28 proteins are composed of a cold-shock domain (CSD) and a zinc knuckle (ZK) domain belonging to the zinc finger CCHC-type (ZCCHC) superfamily (Fig. 1c).

To better understand the uridylation mechanism involved in the biogenesis of mature let-7, we used single-particle cryo-EM to study the structures of TUT7 and TUT4 without and with pre-let-7g miRNA (Fig. 1d, see Supplementary Information for RNA sequences), in the absence or presence of LIN28A. We purified recombinant full-length wild-type human TUT7 and human LIN28A from *Escherichia coli* strain KRX[38]. Synthetic pre-let-7g was used as a substrate for structural studies, because it has been widely used in previous studies of TUT4/7 activity[4,34]. To visualize the TUT7 apo state we initially isolated the enzyme bound with UTPαS alone (Extended Data Fig. 1a,b) but found that modification with NHS-polyethyleneglycol4 (NHS-PEG4)[39] was needed to avert excessive sticking of particles to the air−water interface where they are prone to denaturation (Extended Data Fig. 1b,c). Unmodified enzyme was combined with pre-let-7g to obtain the binary TUT7/pre-miRNA complex (Extended Data Fig. 1b,d) and the TUT7/LIN28A/pre-let-7g ternary complex (Extended Data Fig. 1b,e). The activity of TUT7 as expressed here and assembled with pre-let-7g miRNA was confirmed both without and with LIN28A, resulting in mono- and oligo-uridylation, respectively (Extended Data Fig. 1f). Using the isolated apo form and assembled sample complexes we embedded the TUT7 samples in vitreous ice and solved a series of cryo-EM structures in different functionally related states and with overall resolutions ranging from 3.53 to 4.03 Å (Table 1). To compare with TUT7, a truncated TUT4 (residues 254–1315) with a similar activity to the full-length TUT4 protein and to an N-terminal-only truncated form (residues 153–1649) was expressed and purified from the *E. coli* KRX strain and assembled with LIN28A and pre-let-7g before polishing by size-exclusion chromatography (Extended Data Fig. 1g–i). As before, enzyme activity for the truncated TUT4 construct was confirmed both without and with LIN28A (Extended Data Fig. 1j). We thereby succeeded in obtaining a cryo-EM map of the TUT4 ternary complex at a resolution of 3.68 Å (Table 1).

### Loose association between NTDs characterizes the TUT7 apo state

The TUT7 structure in an apo state at 4.03 Å revealed an overall 'C-shape' conformation composed of two right-handed conformers (Fig. 1e and Extended Data Fig. 2a–f) distinct from the 'L-shape' of human Dicer, which in addition remains similar in its conformation throughout its activity cycle[14]. This difference suits Dicer's role as a 'tape-measure' protein capable of recognizing the length associated with a correctly generated pre-miRNA, hence its need for a fixed dimension, whereas the TUTs need to be able to function in two different modes (mono- and oligo-uridylation) and, further, to be able to act processively on a bound pre-miRNA substrate under the influence of LIN28A.

### TUT7 alone binds pre-let-7g miRNA via end-capture of the target 3′ end

A 3.55 Å map of TUT7 in a binary complex with a pre-let-7g (Fig. 1f,g and Extended Data Figs. 2g–l and 3a) shows the RNA stem bound into the catalytic cleft of its CM, with the cytosine single-nucleotide overhang at the pre-miRNA 3′ end ($C^{78}$) and UTPαS bound and stabilized by the critical residues; F1045, S1047, Q1124, L1126, N1130, K1156, S1170, Y1171 and H1286 (Fig. 1h and Extended Data Fig. 3b) (H1286 is equivalent to H336 in the yeast homolog that is known to be critical for uracil selection[40–42]). The let-7g pre-element is oriented away from the CM and, lacking contact with ordered protein structure, was found to be relatively disordered, presumably due to some intrinsic flexibility. Interpretation of contacts made by TUT7 with the RNA stem was assisted by modeling of its ZK1, which fitted well into a clearly visualized but relatively low-resolution region of density (Fig. 1f and Extended Data Fig. 3c,d). The ZK1 region had not been resolved in the previous structural study of the CM alone[34], which instead provided an atomic model for ZK2, a region of TUT7 that supports oligo-uridylation by engaging the extending U tail[34], but is not represented in our structure due to disorder.

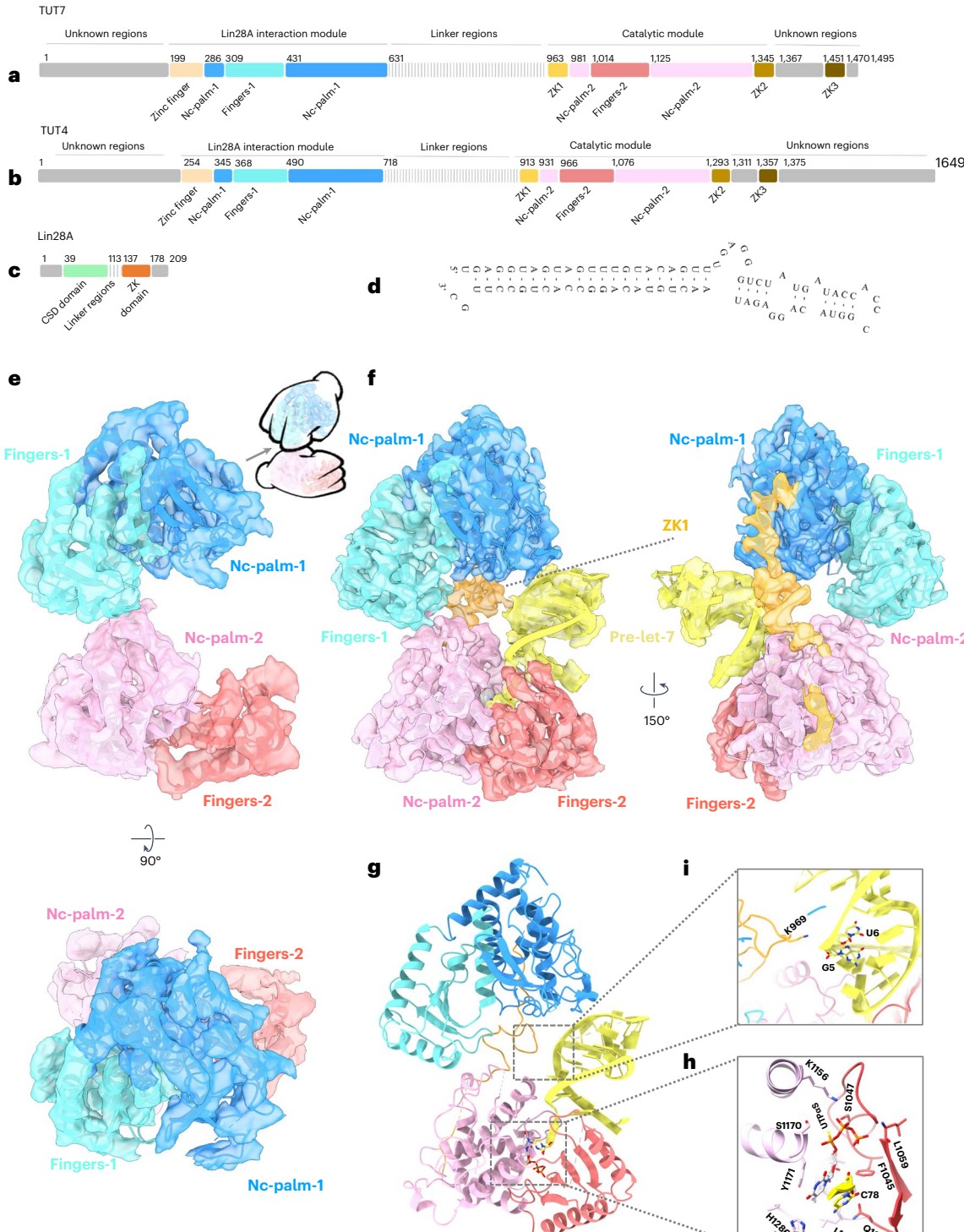

**Fig. 1 | Cryo-EM structures of TUT7 in apo and binary complex with pre-let-7.**
**a**, Human TUT7 domain organization. **b**, Human TUT4 domain organization.
**c**, Human LIN28A domain organization. **d**, Nucleotide sequence of pre-let-7g,
presented in a predicted secondary structure. **e**, Overall structure of TUT7
alone, shown as a 4.03 Å cryo-EM density map with cartoon representation of
protein backbone, in two views related by a 90° rotation, as indicated. A handed
model of TUT7 apo is presented at the upper right corner. The canonical palm
and finger domains of the NTD canonical fold are colored as follows, N-terminal
pseudo-NTD (from LIM) in red and pink and C-terminal NTD (CM) in blue and
cyan. **f**, Overall structure of the TUT7/pre-let-7g binary complex, shown as a
3.55 Å cryo-EM density map with cartoon representation of the fitted coordinates
in two views related by a 150° rotation, as indicated. TUT7 colored as in **e**, ZK1
colored orange, pre-let-7g yellow. **g**, Cartoon model of TUT7 in complex with
pre-let-7g, highlighting the relative arrangements of domains in a C-shape with
the pre-miRNA bound. **h**, Zoomed-in view of UTPαS within the catalytic pocket,
interacting with the terminal pre-let-7g base C[78] and the TUT7 residues shown.
**i**, Close-up of RNA binding by ZK1, highlighting the role of the K969 side chain
interacting with G[5] and U[6] within the pre-let-7g stem.

In both the apo state (Fig. 1e) and the binary TUT7 complex
(Fig. 1f,g) the two NTDs maintain the distinctive 'C-shape', apparently
due to their direct interaction with one another. The interface between
the LIM and CM is occupied by several nonpolar and polar amino
acids, indicating the likelihood of a hydrophobic lateral interaction,
as observed also in Dicer through the RNase IIIa and IIIb domains[14].

**Table 1 | Cryo-EM data collection, refinement and validation statistics**

| | TUT4/RNA/Lin28A | TUT7 apo | TUT7/RNA | TUT7/RNA/Lin28A-1 | TUT7/RNA/Lin28A-2 |
|---|---|---|---|---|---|
| | (EMD-17164), (PDB 8OST) | (EMD-16825), (PDB 8OEF) | (EMD-17084), (PDB 8OPP) | (EMD-17086), (PDB 8OPS) | (EMD-17087), (PDB 8OPT) |
| **Data collection and processing** | | | | | |
| Magnification | 105,000 | 105,000 | 130,000 | 130,000 | 130,000 |
| Voltage (kV) | 300 | 300 | 300 | 300 | 300 |
| Electron exposure (e$^-$/Å$^2$) | 50 | 50 | 60 | 50 | 50 |
| Defocus range (µm) | 1.5–3 | 1.5–3 | 1.5–3 | 1.5–3 | 1.5–3 |
| Pixel size (Å) | 0.83 | 0.83 | 0.93 | 0.93 | 0.93 |
| Symmetry imposed | C1 | C1 | C1 | C1 | C1 |
| Initial particles images (no.) | 1,815,635 | 1,415,092 | 1,025,792 | 1,875,928 | 1,875,928 |
| Final particles images (no.) | 266,487 | 200,865 | 186,567 | 169,985 | 180,310 |
| Map resolution (Å) | 3.68 | 4.03 | 3.55 | 3.81 | 3.53 |
| FSC threshold | 0.143 | 0.143 | 0.143 | 0.143 | 0.143 |
| Map resolution range (Å) | 3.6–8 | 4.1–10 | 3.5–8 | 3.8–8 | 3.5–8 |
| **Refinement** | | | | | |
| Initial model used (PDB code) | 6IW6/3TS2 | 5WOB | 5WOB | 5WOB/3TS2 | 5WOB/3TS2 |
| Model resolution (Å) | 3.7 | 4.1 | 3.6 | 3.8 | 3.5 |
| FSC threshold | 0.143 | 0.143 | 0.143 | 0.143 | 0.143 |
| Model resolution range (Å) | 3.6–7.2 | 4.1–8.1 | 3.5–6 | 3.8–7.4 | 3.5–4 |
| Map sharpening $B$ factor (Å$^2$) | 156.5 | 237.4 | 128.9 | 155.3 | 103.7 |
| **Model composition** | | | | | |
| Nonhydrogen atoms | 9,007 | 9,687 | 11,685 | 9,304 | 14,968 |
| Protein residues | 432 | 612 | 670 | 445 | 825 |
| Ligands (Zn$^{2+}$/UTPαS) | Zn$^{2+}$:2 | 0 | UTPαS:1 | Zn$^{2+}$:1 | Zn$^{2+}$:2 |
| **$B$ factors (Å$^2$)** | | | | | |
| **Protein** | | | | | |
| Ligand (nucleotides) | 69 | 0 | 25 | 69 | 54 |
| **Root mean squared deviations** | | | | | |
| Bond lengths (Å) | 0.002 | 0.002 | 0.002 | 0.003 | 0.002 |
| Bond angles (°) | 0.511 | 0.472 | 0.453 | 0.568 | 0.476 |
| **Validation** | | | | | |
| MolProbity score | 1.26 | 1.31 | 1.28 | 1.35 | 1.46 |
| Clashscore | 4.96 | 5.68 | 5.23 | 6.28 | 6.94 |
| Poor rotamers (%) | 0 | 0 | 0 | 0.78 | 0.14 |
| **Ramachandran plot** | | | | | |
| Favored (%) | 98.82 | 98.31 | 98.01 | 99.31 | 97.65 |
| Allowed (%) | 1.18 | 1.69 | 1.99 | 0.69 | 2.35 |
| Disallowed (%) | 0 | 0 | 0 | 0 | 0 |

The binding of the pre-miRNA results in a higher-resolution structure with more of TUT7 properly resolved in the cryo-EM density, and with an extensively extended polypeptide region linking the LIM and CM that includes its ZK1 region (Extended Data Fig. 3d). Thus partially stabilized, the binary TUT7/pre-miRNA complex shows a defined but dynamic interface between the LIM and CM domains (Supplementary Video 1). By contrast, in the apo state no obvious density was resolved for the backbone link connecting the LIM and CM, with only a small region of the linking density visible associated with the LIM (Extended Data Fig. 3e). Nevertheless, for maintenance of the distinctive 'C-shape' structure, the two NTDs of the enzyme must still be interacting directly in the apo state, albeit relatively weakly. These two models show us

the conformation of TUT7 (and we posit, TUT4) in the state before substrate binding (Fig. 1e) and when a pre-miRNA alone associates with it (Fig. 1f,g).

The roles of all three TUT4/7 ZKs were probed in a recent NMR-based study that made use of pre-let-7i as a target[43]. While binding of TUT4 ZK2 and ZK3 to RNA was shown, it was concluded that ZK1 had lost its ability to interact with RNA due to two key mutations within its CCHC motif after the first cysteine and the single histidine, where the relevant amino acids were mutated from a variable bulky mostly hydrophobic residue to a serine[43]. Alignment of the amino acids of the three ZKs of TUT7 (Extended Data Fig. 3f) showed the residue after the first cysteine of ZK1 was serine, but the one after its histidine

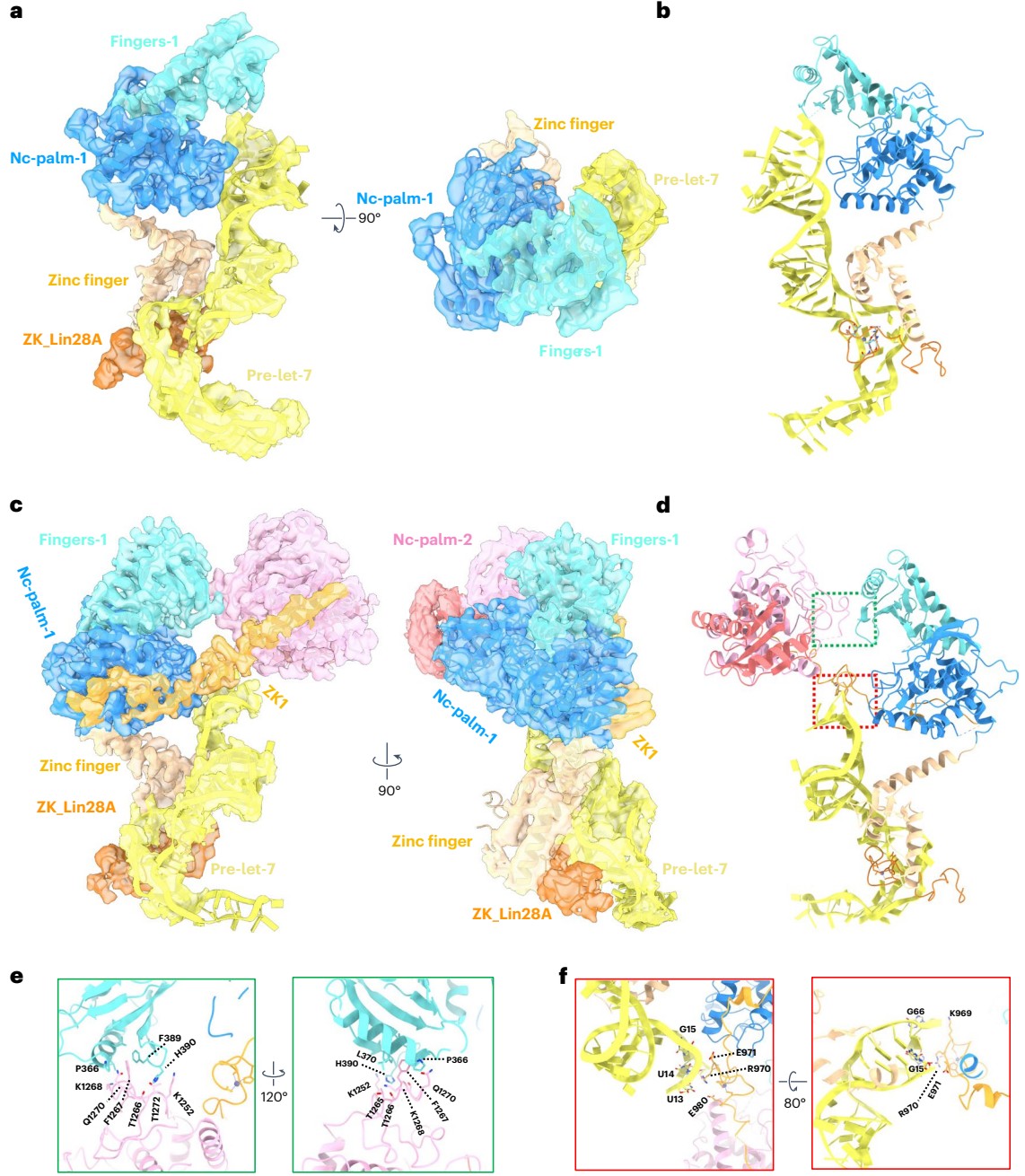

**Fig. 2 | Cryo-EM structures of TUT7 in complex with pre-let-7g and LIN28A.**
**a**, Overall structure of the TUT7/pre-let-7g/LIN28A ternary complex conformation I, shown as a 3.81 Å cryo-EM density map and cartoon representations. C-terminal NTD (CM) fingers in cyan and palm in blue; the LIM zinc finger is colored sand, LIN28A ZK dark orange and pre-let-7g miRNA yellow, in two views related by a 90° rotation. **b**, Cartoon model of TUT7/pre-let-7g/LIN28A ternary complex conformation I, colors as in **a**. **c**, Overall structure of the TUT7/pre-let-7g/LIN28A ternary complex conformation II, shown as a 3.53 Å cryo-EM density map and cartoon representation. Colors as in **a** and **b**, plus TUT7 ZK1 and linker region in bright orange, and pseudo-NTD from the LIM (fingers red and palm pink), in two views related by a 90° rotation as indicated. **d**, Cartoon model of TUT7/pre-let-7/LIN28A ternary conformation II. Areas shown in expanded views below are boxed. **e**, Zoomed-in, two views of domain interactionn between TUT7 LIM and TUT7 CM related by a 120° rotation as indicated. **f**, Close-up of TUT7 ZK1 bound to the pre-let-7, highlighting key roles for K969 and for K970, E971 and E980, among other residues in two views related by an 80° rotation.

was leucine, different from the situation in TUT4 where it is another serine[43]. Nevertheless, in the binary complex we resolve, the lysine (K969) after the second TUT7 cysteine of ZK1 is shown to interact with the G[5] and U[6] of pre-let-7 using its side chain (Fig. 1i and Extended Data Fig. 3c). This reveals that one of the functions for ZK1, at least in TUT7, is to recruit or stabilize pre-let-7 for mono-uridylation through a direct interaction with its linker loop. K969 of TUT7 is conserved in the ZK1 of TUT4 as K919 (Extended Data Fig. 3f).

**LIN28A modifies target pre-miRNA recognition by previous association**

Two distinct conformations (I and II) of the TUT7 ternary complex were resolved at 3.81 and 3.53 Å resolutions, respectively (see Extended Data Fig. 4 for the data processing pipeline and Extended Data Fig. 5 for close-ups of local density). In conformation I (Extended Data Figs. 4a–f and 5a), most of the pre-let-7g could be resolved (Fig. 2a,b). In this conformation, the RNA stem interacts with the TUT7 LIM, while the

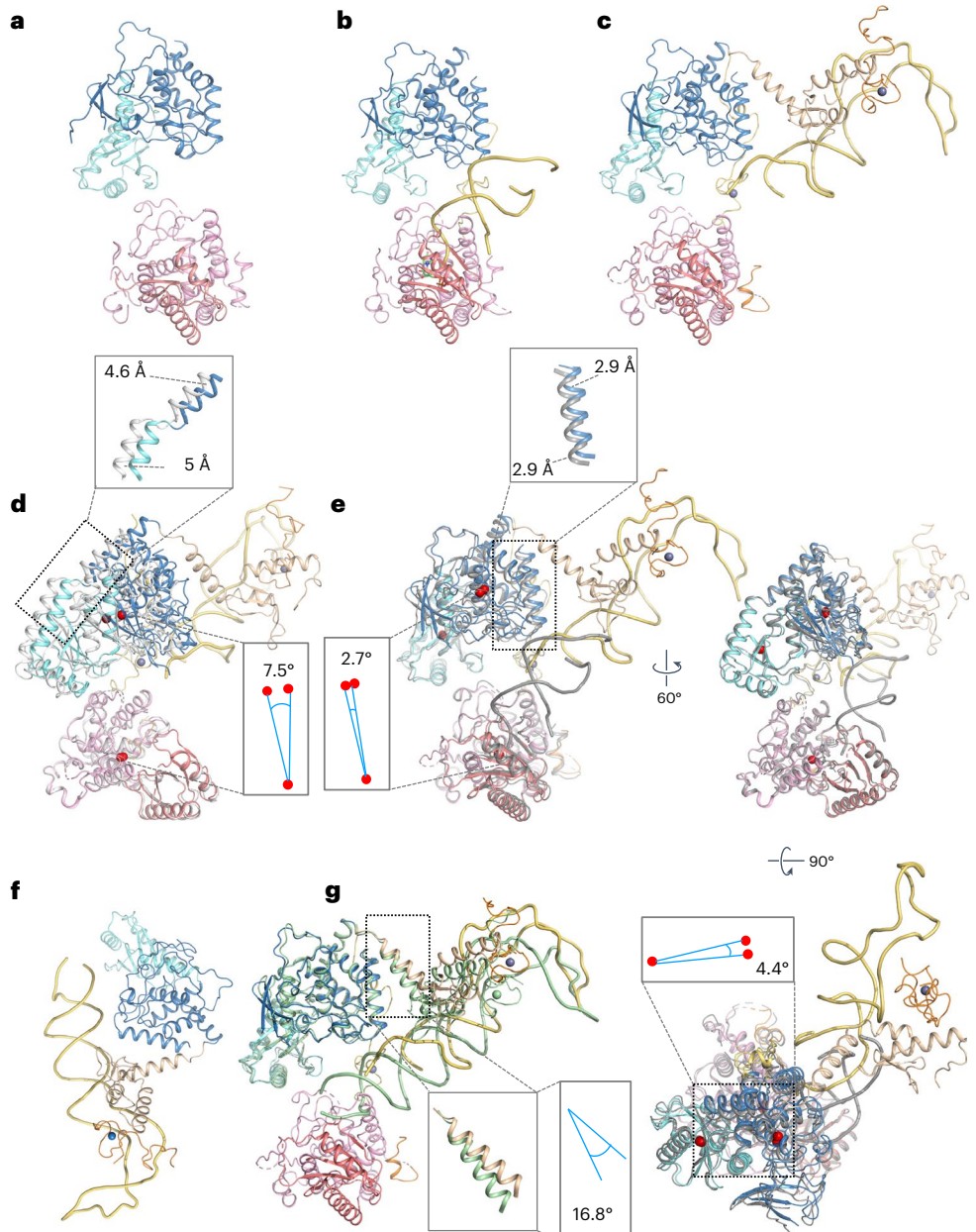

**Fig. 3 | Structural comparison of TUT7 cryo-EM structures. a–c,** Cartoon models of TUT7 in apo state (**a**), as a binary complex with pre-let-7g (**b**) and in ternary complex with pre-let-7g and LIN28A (conformation II) (**c**). Domains are colored as in the main figures and zinc ions are blue spheres. **d,** Superimposition of the TUT7 apo (white) and pre-let-7g bound cryo-EM structure from the TUT7/pre-let-7g/LIN28A ternary complex conformation II (colored), aligning by the TUT7 CM. The reorientation of the LIM results in a ~5 Å shift toward the pre-miRNA, as shown. Angular changes in this figure were calculated by reference to the centers of mass of the respective regions, marked here by red dots. **e,** Superimposition of the TUT7/pre-let-7g binary complex (deep gray) and the TUT7/pre-let-7g/LIN28A complex conformation II (colored), in two views related by 60° and 90° as shown, and aligning by the TUT7 CM. **f,** Cartoon model of TUT7/pre-let-7g/LIN28A ternary complex conformation I. **g,** Superimposition of the TUT7/pre-let-7g/LIN28A complex conformation I (pale green) and conformation II (colored), aligning by the TUT7 LIM.

let-7g [49]AGGAG[53] region coordinates with the zinc finger of TUT7 and ZK domain of LIN28A, forming a stable ternary complex[35] (Fig. 2a,b). Furthermore, despite clear density representing the pre-element (the apical loop and short stem region preceding later removed by Dicer action), no density was observed for the CSD of LIN28A, presumably due to the relative flexibility between the LIM + LIN28A and pre-let-7g. Based on the position of the RNA, this complex is believed to represent the initial step in oligo-uridylation, a 'capture' state in which the pre-let-7 has begun association with TUT7 but not yet completed its engagement.

In conformation II (Fig. 2c,d and Extended Data Figs. 4a–c,g–i and 5b), the CM density missing in conformation I due to intrinsic flexibility could be entirely resolved as well as the LIM-CM linker region including the ZK1 domain (Extended Data Fig. 6a). The higher-resolution model enabled us to observe intramolecular interactions at a residue level, especially for the interface between the LIM and CM. For example, a hydrophobic interaction formed by ring-stacked residues F389 and F1267, and a H390-induced polar interaction with T1266 and T1272 stabilize the interface (Fig. 2e and Extended Data Fig. 6b), thus also confirming the role of a similar interface in the TUT7 binary complex (Fig. 1f), and presumably the apo state also. Together, these interactions ensure maintenance of the distinctive conformation of TUT4/7. In this state of the ternary complex R970 forms two hydrogen bonds

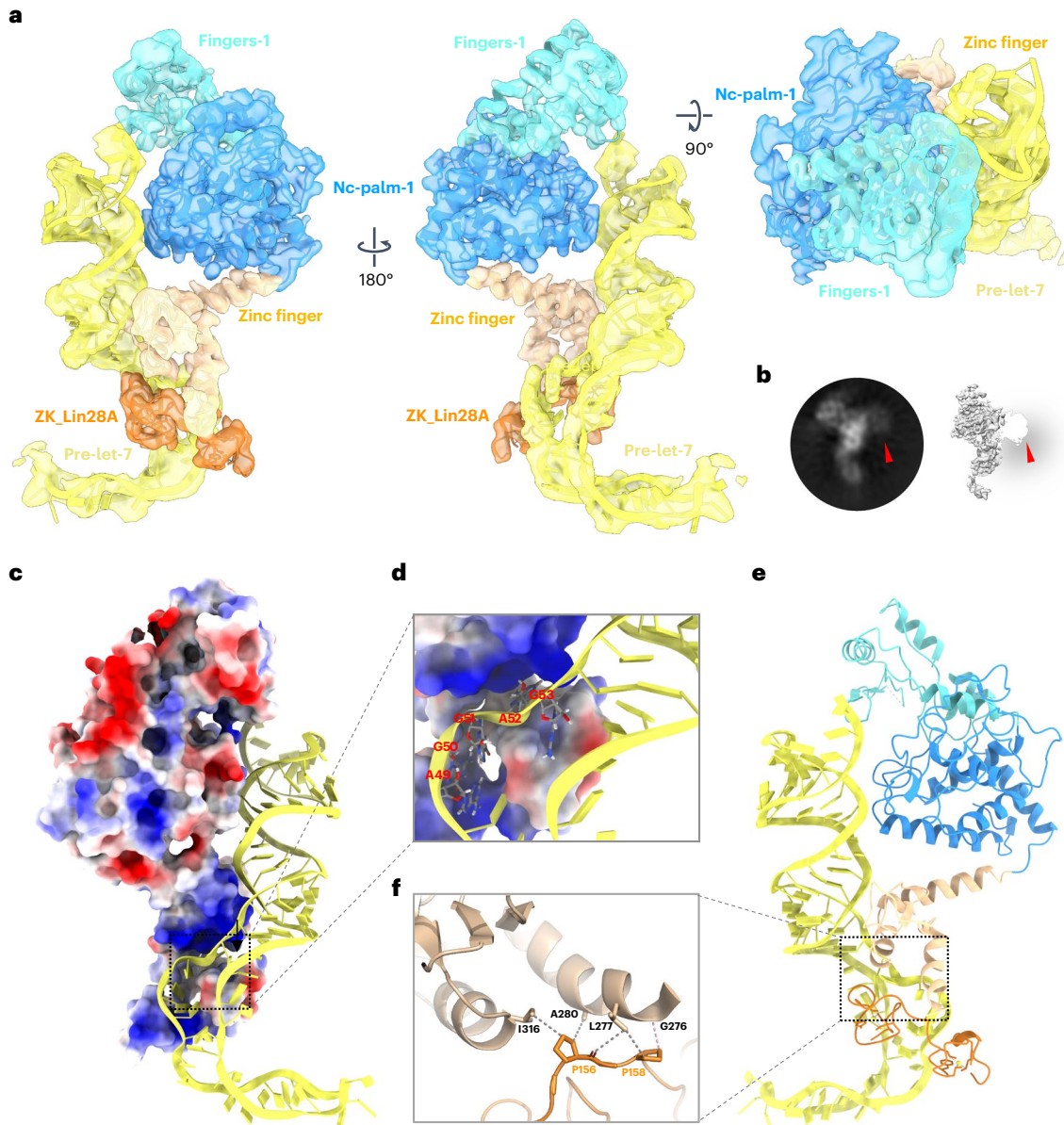

**Fig. 4 | Cryo-EM structure of TUT4 in complex with pre-let-7g with LIN28A.**
**a**, Overall structure of TUT4/pre-let-7g/LIN28A, shown as a 3.68 Å cryo-EM density map and cartoon representations in three views, related by 90° and 180° rotations, as shown. Colors are as previously: pseudo-NTD (LIM) fingers and palm in cyan and blue, respectively; LIM zinc finger and linking density toward CM is colored sand, pre-let-7g yellow and LIN28A ZK dark orange. **b**, A typical 2D class average representing the poorly resolved density from the CM (marked by a red arrowhead). **c**, Electrostatic surface potential of TUT4/pre-let-7g/LIN28A.

The positively and negatively charged regions are colored from blue to red, respectively. **d**, Zoomed-in view of the pre-let-7g [49]AGGAG[53] region in a positively charged pocket created by the TUT4 zinc finger and ZK domain of LIN28A (black arrows). **e**, Cartoon model of TUT4/pre-let-7g/LIN28A ternary complex, colored as in other panels. **f**, Zoomed-in view of TUT4/LIN28A interface. Gray dashed lines link atoms within 3.8 Å and the pale pink dashed line shows a distance (at G276) between two atoms of 3.8–4 Å.

with $U^{13}$ and $U^{14}$ in the RNA stem while K969 binds the sugar backbone of $G^{66}$ (Fig. 2f and Extended Data Fig. 6c), which is different from the binary complex in which its side chain instead interacts with $G^5$ and $U^6$ (Extended Data Fig. 3c). In addition, residues E971 and E980 from ZK1 further stabilize RNA binding by interactions with the pre-let-7g backbone ($G^{15}$) and the base of $U^{13}$ (Fig. 2f). Altogether, these interactions aid the translocation of pre-let-7 to close the catalytic pocket of the CM and stabilize the TUT7 bound to the pre-miRNA and LIN28A.

Both models of ternary complexes revealed the ZK domain of LIN28A attached to its [49]AGGAG[53] recognition sequence in the pre-let-7 together with the zinc finger of the TUT7 LIM. To explore the potential conformational changes in the overall structure, a three-dimensional

(3D) conformational variability analysis was performed (Supplementary Video 2). The TUT7 LIM and LIN28A were found to be stable, while the pre-element of pre-let-7g miRNA exhibited density that could be assigned to a set of flexibly related conformations. The angle between the pre-element and RNA stem changed from around the initial (frame no. 1) 70° to a final (frame no. 20) 40°. The finding was repeated via the same analysis for the ternary complex in conformation II (Supplementary Video 3). This result is consistent with a previous study, in which it was concluded that in solution, the pre-let-7 alone may adopt different conformations with various stem stabilities[14], which we now observe in our cryo-EM structures bound to TUT7. Since the CSD was thought to bind in the pre-element region this may explain why this density could not be resolved.

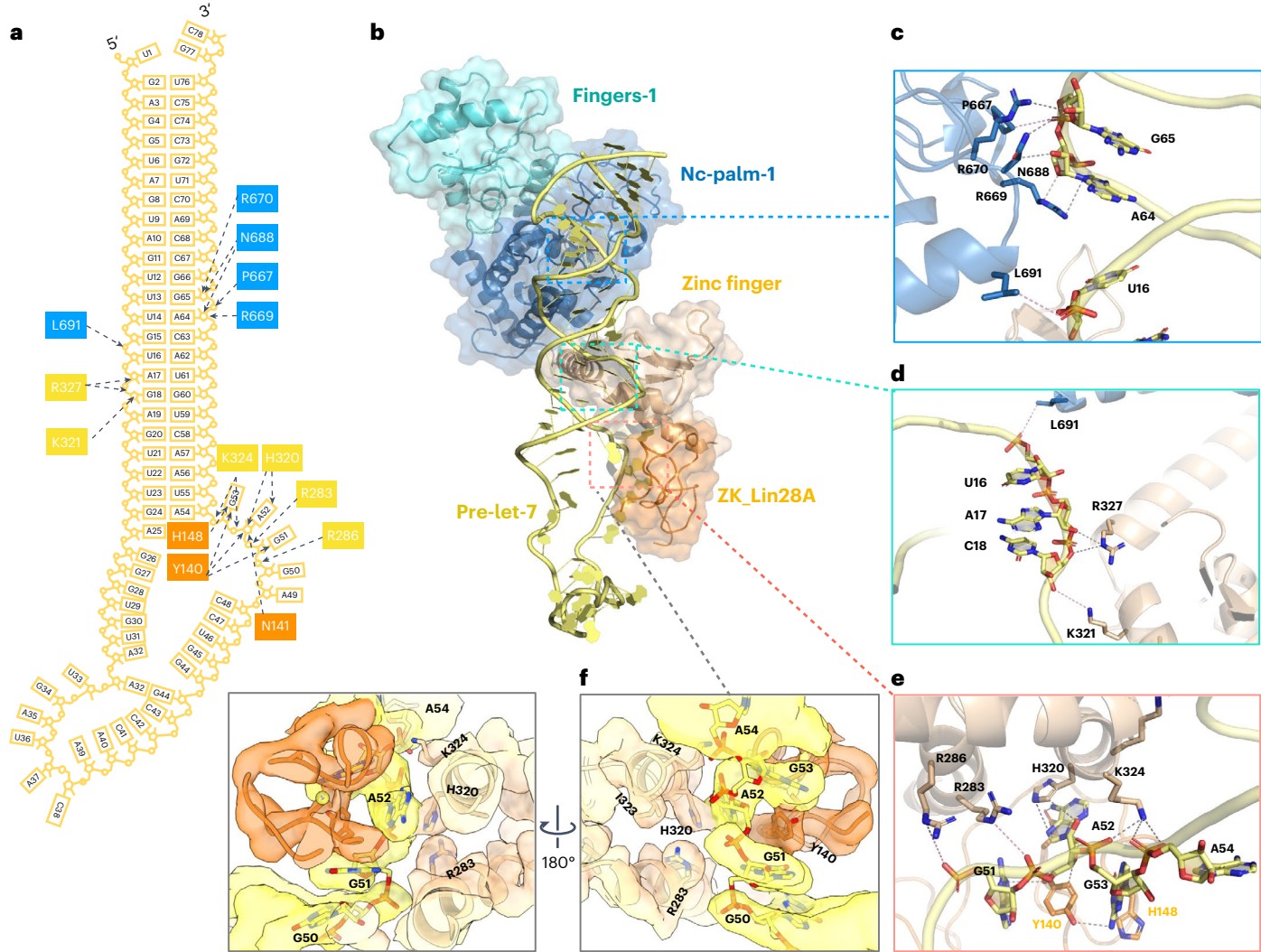

**Fig. 5 | Pre-let-7 recognition by TUT4 and LIN28A. a**, A schematic of pre-let-7 recognition by TUT4 and LIN28A, based on the structure reported here (also Fig. 4). **b**, Structure of TUT4/LIN28A bound to pre-let-7g. **c–f**, Zoomed-in views of the different interfaces bound to pre-let-7g: interface 1 (**c**), interface 2 (**d**) and interface 3 (**e** and **f**). Throughout this figure, gray dashed lines link atoms within 3.8 Å and pale pink dashed lines show a distance between two atoms of 3.8–4 Å.

Altogether, comparison of the two TUT7 ternary complexes and the binary and apo structures (Fig. 3a–g) indicates that the presence of LIN28A has changed how the pre-let-7 engages with the enzyme, directing it away from direct binding to the CM active site (as in the binary complex) and toward a two-stage process in which, first, the LIM and LIN28A capture the stem and pre-element, and then the CM engages with the 3′ end. Comparison of the apo TUT7 and its ternary complex conformation II indicates a 7.5° rotation of the LIM region with respect to the CM, while the switch from binary TUT7 to ternary conformation II involves less than half of this movement (2.7°); additionally, the Nc-palm-1 moves 4.4° with respect to the Fingers-1 subdomain, within the LIM, as the TUT7 binary complex converts ultimately to ternary conformation II (Fig. 3d,e). By contrast, comparison of TUT7 ternary conformations I and II finds no change in the relative positions of the palm and fingers subdomains of the LIM while the zinc finger binding the pre-let-7 stem moves substantially about a 16.8° rotation of the linking helix (Fig. 3f,g). Further details of the first of these steps—formation of the capture complex—were provided by determination of the equivalent structure for TUT4, also with LIN28A and pre-let-7g.

## TUT4 capture complex determined at higher resolution
A 3.68 Å cryo-EM structure was obtained for the TUT4/LIN28A/pre-let-7 in the pre-oligo-uridylation capture state (Extended Data

Fig. 7) found also as TUT7 ternary conformation I. This structure enabled us to identify more clearly the interactions between the LIM module of TUT4, LIN28A and the pre-let-7 (Fig. 4a). Although the CM of TUT4 is not well defined in the 3D map, density apparently derived from it can be visualized in two-dimensional (2D) class averages of the complex, roughly oriented in the same view compared to the TUT4 ternary complex in conformation I (Fig. 4b), in line with this being a capture state before CM engagement with the open end of the pre-let-7g. The [49]AGGAG[53] recognition sequence from pre-let-7g was observed trapped into a positively charged pocket created by TUT4 and LIN28A (Fig. 4c,d). Together with the TUT7 ternary complexes this structure therefore provides a demonstration that TUT4 interacts directly with LIN28A, a currently unresolved and key question in the field. In this ternary interface, a hydrophobic patch is formed between G276, L277, A280, I316 from TUT4 LIM (which are conserved in TUT7: G216, L217, A220, I256) and prolines 156 and 168 of LIN28A (Fig. 4e,f). Overall, the structures of the TUT4 and TUT7 LIMs are very similar (root mean squared deviation 1.22 Å over 302 core backbone Cα atoms, 2.18 Å over all 380 pairs) (Extended Data Fig. 8a,b). Compared to the crystal structure of TUT4 LIM[33] (Extended Data Fig. 8c), the angle between the helices α4 in the zinc finger and α5 in the palm subdomain is reduced by 12.6° during the binding of pre-let-7g (Extended Data Fig. 8d).

The pre-let-7g is observed bound to three main interfaces in the TUT4 capture complex (Fig. 5a), which we have termed interface 1, interface 2 and interface 3, respectively (Fig. 5b–f). In interface 1, the TUT4 LIM P667, R669, R670 and N688 interact with A$^{64}$ and G$^{65}$ at their deoxyribose and phosphate group (Fig. 5c). The pre-let-7 A$^{17}$ and C$^{18}$ backbone regions are bound by R327 and K321, respectively, in the second interface region, while the U$^{16}$ backbone interacts with L691 (Fig. 5d). Interface 3 is critical to the ternary interaction: in this region the zinc finger of the TUT4 LIM is observed to bind the $^{49}$AGGAG$^{53}$ region of pre-let-7 and the two ZKs of LIN28A. The side chains of four positively charged residues R283, R286, H320 and K324 from the LIM participate in the interaction with G$^{51}$, A$^{52}$, G$^{53}$ and A$^{54}$ of the pre-let-7g (Fig. 5e,f).

The two ZKs of LIN28A interact with the $^{49}$AGGAG$^{53}$ motif of pre-let-7g, in a similar way to the previously solved isolated complexes[35,44]. Based on these structures, it was concluded that the double ZK motif recognizes two AG dinucleotides separated by a single-nucleotide spacer[35,44]. Perhaps due to the use of a modified RNA sequence mimicking the full pre-let-7g, the exact basis for recognition observed differed from the one we identify. In the previously solved crystal structures the LIN28A ZK1 binds with the A$^{54}$ of pre-let-7g via a hydrogen bond from the Y140 sidechain, whereas we observe the tyrosine binding with G$^{51}$, A$^{52}$ and G$^{53}$ (Fig. 5a,e). In all, the pre-let-7 miRNA is sandwiched between LIN28A and the TUT4 LIM (Figs. 4c,d and 5b–f and Extended Data Fig. 8e) as it was in the TUT7/pre-let-7g/LIN28A complex (Extended Data Fig. 8f,g) and so as to engage with the pre-miRNA in a fundamentally different way from that observed for the binary TUT7/ pre-let-7g complex set for mono-uridylation (Extended Data Fig. 8h). The binding mode and overall conformation of the pre-let-7g miRNA appears to be remarkably consistent in all three LIN28A-containing complexes (Extended Data Fig. 8e–g), apparently due to the LIM/ LIN28A clasp (Fig. 5f) and other key interfaces shown by mutational data to be the specific interactions responsible for ensuring efficient and processive oligo-uridylation.

Mutagenesis of TUT4 (Fig. 6a) in interfaces 2 and 3 still enabled addition of a single uridine to pre-let-7g in the absence of LIN28A (Fig. 6b), but abolished the enzyme's capacity to oligo-uridylate pre-let-7 in the presence of LIN28A (Fig. 6c–e) due to a much-reduced ability to form a TUT/pre-let-7g/LIN28A ternary assembly (Fig. 6f,g). Although mutations in the interface 1 affected the formation of the ternary complex (Fig. 6f,g), they did so less than interface 2 and 3 mutations, and still permitted oligo-uridylation, albeit with a lower efficiency compared to wild type (Fig. 6c–e). No single-site mutations had a notable impact on TUT4 activity, hence the use of double mutants in the assays shown.

To our surprise, mutations in the ZK1 region that binds to the RNA stem in the TUT7 ternary conformation II, had no effect on oligo-uridylation activity (Fig. 6c–e), indicating other residues play critical roles in RNA binding, for example D921 and E930 that are the equivalent residues to TUT7 E971 and E980 (Fig. 2f and Extended Data Fig. 9). It is notable that the key interactions we observed in TUT4 complexes involve residues conserved in TUT7 (for example, K919/K920 in TUT4 equivalent to K969/R970 in TUT7), with the one notable difference

that L691 in human TUT4 is P613 in TUT7, and is a polypeptide locus in interface 1 besides U$^{16}$ of pre-let-7g; and otherwise only the conservative substitution of R669 in human TUT4, for K591 in TUT7 (Extended Data Fig. 9). Higher-order assemblies of TUT4/pre-let-7g/LIN28A can also be observed in the shift assays testing complex assembly, for wild-type TUT4 and with mutations at interface 1 and ZK1 (Fig. 6f,g), and we confirmed a concentration-dependent formation of these larger complexes by titrating pre-let-7g with LIN28A and TUT4 alone and together (Fig. 6h).

## Discussion

Together the structures we have determined show that LIN28A induces a large conformational change, including of the CM and the Zinc finger in the LIM, on engagement with pre-let-7g miRNA and TUT4/7. After binding to pre-let-7g/LIN28A, the zinc finger in the TUT7 LIM was observed with good density, whereas it was unobserved in TUT7 apo or binary states because of mobility and/or disorder (Fig. 3a–c,f). In the TUT4 ternary complex similarly the LIM was well-resolved in complex with LIN28A and pre-let-7g miRNA (Extended Data Fig. 8a). Where pre-let-7 miRNA was shown to be flexible it was previously found to form a fixed conformation within the context of a ternary Dicer complex incorporating TRBP[14]; similarly we observe a well-ordered pre-let-7g conformation after TUT4/7 binding alongside LIN28A. Overall, the role of LIN28A appears to be to (1) directly interact with the pre-let-7 miRNA; (2) to enable the LIM zinc finger to also bind to the pre-let-7 RNA (since this interaction is absent in the TUT7 binary complex) and (3) to directly interact with the TUT LIM domain as well. The structures of TUT7 and TUT4 bound to pre-let-7g miRNA and to LIN28A allow us to propose a mechanistic model for mono-uridylation and for oligo-uridylation, as follows (Fig. 7).

In the mono-uridylation process, the TUT7 alone presents a pre-state (Figs. 1e and 7a(i)). Next, TUT7(4) recruits the pre-let-7, binding its open stem (binary complex, Fig. 1f,g). The catalytic pocket engages with the 3′ end of the RNA, while the ZK1 binds to the RNA stem via key residue K969 (TUT7; K919 TUT4; Fig. 1i). This stabilized state aids the addition of a single uridine, indicating the binary complex we observe as an engaged active state (Fig. 7a(ii)). Finally, the mono-uridylated pre-let-7 becomes unstable, which subsequently leads to the release of product (Fig. 7a(iii)). In the presence of LIN28A, the potential order of events during pre-let-7 capture and engagement changes and there are different pathways to ternary complex formation. Either the pre-let-7 might first associate with LIN28A and then recruit the TUT enzyme via binding to the LIM (Figs. 2a,b, 4, 5, 6 and 7b(i),(ii), top), to form a capture complex (Figs. 2a,b and 7b(iii)) leading to CM engagement on the open pre-miRNA stem (Figs. 2c–f and 7b(iv),(v)). Or, the preformed binary complex (TUT+pre-miRNA; Fig. 1f,g) might have its pre-let-7 bound and remodeled by the LIN28A (Fig. 7b(ii), bottom) in an alternative sequence of events that still leads to successful ternary complex formation, again initially in a capture state (Fig. 7b(iii)) and ultimately with CM engagement on the open pre-miRNA stem (Fig. 7b(iv),(v)). Affinity measurements recorded in the literature for the Lin28A/pre-miRNA interaction vary over orders

**Fig. 6 | Effects of mutants at the TUT4/pre-let-7g interface on uridylation activity. a**, SDS–PAGE of recombinant TUT4 wild type and its mutants. This expression trial was performed twice. **b**, In vitro uridylation of pre-let-7g by wild-type TUT4 and mutants, in the absence of LIN28A, as indicated. Pre-let-7, substrate, indicated as Marker1, 78 nt, shown in lane 2 (see Supplementary Information for RNA sequence, pre-let-7g); pre-let-7-U, synthetic substrate, indicated as Marker2, 79 nt, shown in lane 1 (see Supplementary Information for RNA sequence pre-let-7g_1U); Mono-U pre-let-7g, pre-let-7g products with a single U addition; Oligo-U pre-let-7g, pre-let-7g products with more than two U addition. K919 and K920 mutated here are the equivalent residues to K969 and R970 in TUT7 (Fig. 2f). This assay was performed three times. **c**, Relative oligo-uridylation activity of wild type or the indicated mutant TUT4 enzymes in the

presence of LIN28A. Markers as for **b**. This assay was performed three times. **d**, In vitro uridylation of pre-let-7g by TUT4 and its variants in the presence of LIN28A. Error bars show the standard error of mean, $n = 3$ independent experiments (trials 1, 2 and 3) (see Source data for the raw data plotted). **e**, Time course activity assay for wild-type and mutant forms of TUT4 in the presence of LIN28A; Marker1, pre-let-7, 78 nt, shown in lane 1. Two repeats were performed for this assay. **f**, EMSA of pre-let-7g with wild-type and mutant forms of TUT4 only. This assay was performed twice. **g**, EMSA of pre-let-7g with wild-type and mutant forms of TUT4, combined with LIN28A and TUT4. This assay was performed twice. **h**, EMSA of pre-let-7g with increasing concentrations of LIN28A alone, TUT4 alone and LIN28A + TUT4 together. Marker1, pre-let-7, 78 nt, 25 kDa, shown in lane 1. This assay was performed twice.

of magnitude, from 0.15 nM to 15 µM (refs. 35,45–48). Nevertheless, data show that the order of addition of components does not influence the ultimate complex formed, since whether all three ternary components are mixed together (Extended Data Fig. 1h) or LIN28A is added to pre-let-7g before addition of TUT4 (Extended Data Fig. 10a–c) or a preformed binary complex has LIN28A added to it (Extended Data

Fig. 10d,e), the outcome is the same, as verified by both size-exclusion chromatography traces and inspection of 2D class averages.

In any case, by altering the mode of binding between the TUT and the pre-miRNA, the intervention of LIN28A enables a much more extensive engagement with pre-miRNAs than in its absence. By defining different ways in which TUT4/7 interact with pre-let-7 miRNAs in

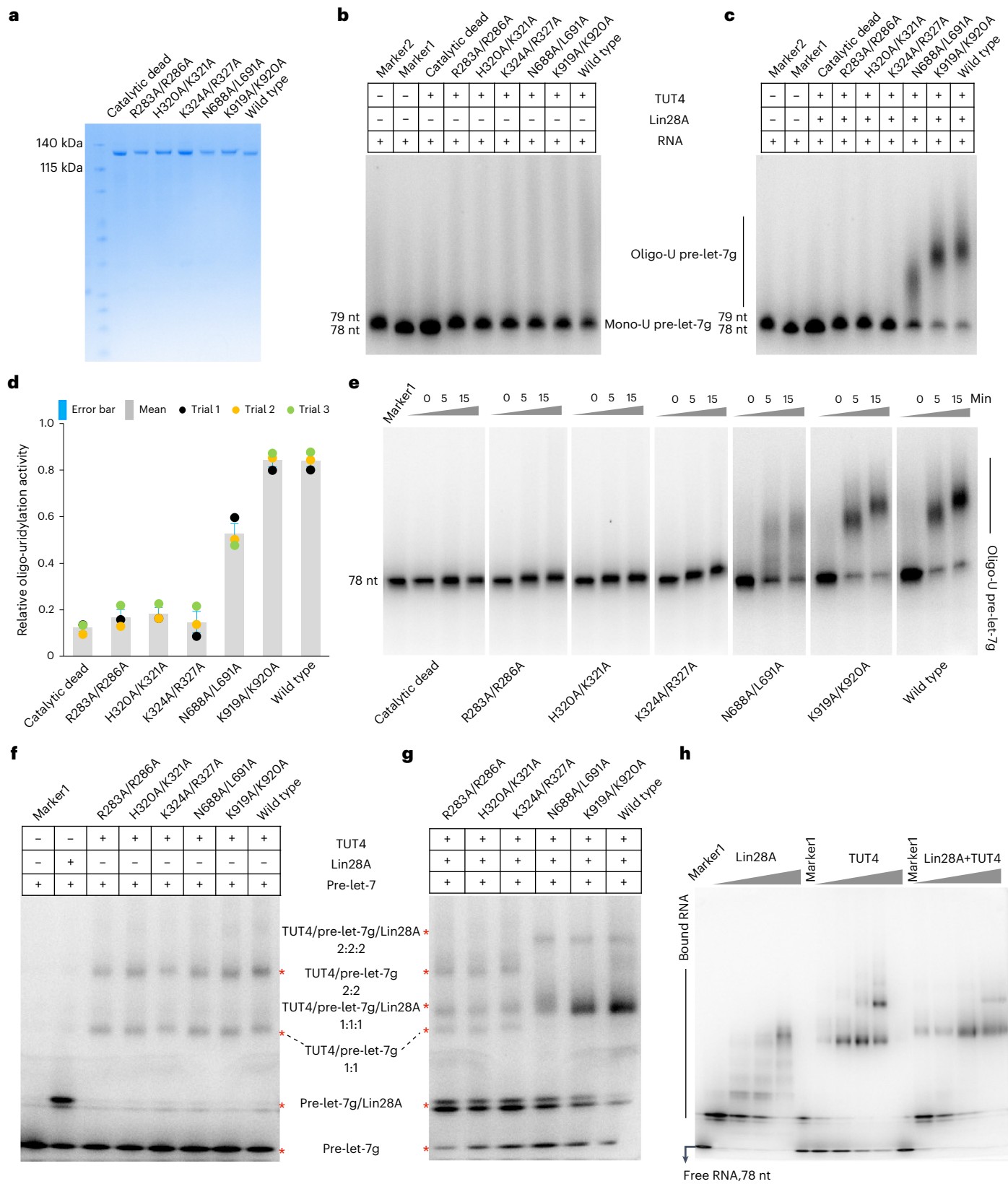

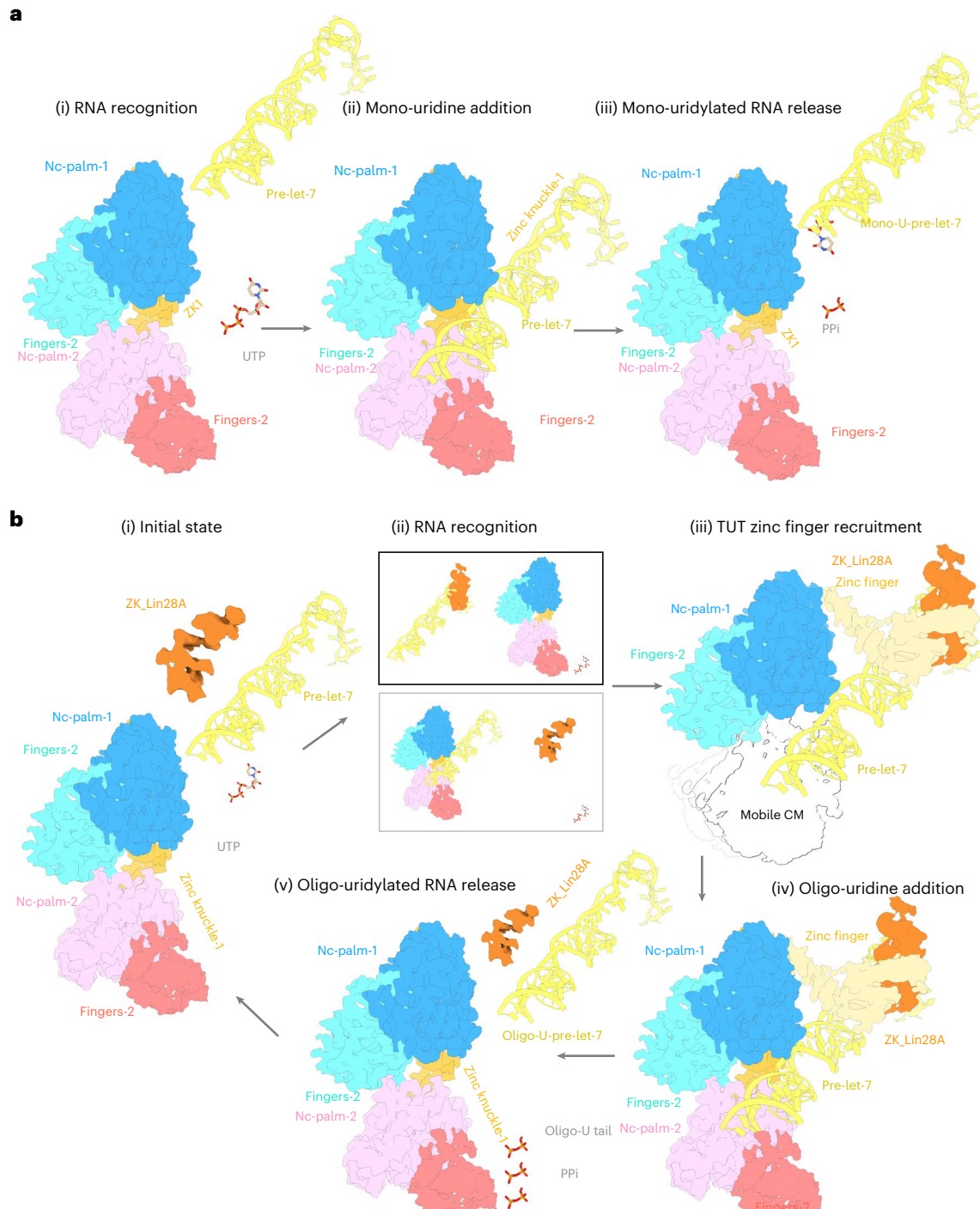

**Fig. 7 | Proposed models of mono-uridylation and oligo-uridylation. a**, (i) The pre-let-7 is initially recognized by apo TUT 4 or 7 and subsequently the substrate is mono-uridylated to add a single uridine-monophosphate to its 3′ end, resulting in a mature Class II pre-let-7 miRNA. (ii),(iii) The mono-uridylated pre-let-7 is then released due to the distributive activity of TUT4/7 in the absence of LIN28A. **b**, (i),(ii) In pre-state-0, the pre-let-7 is initially recognized by LIN28A; (ii) (top) or by TUT4/7; (ii) (bottom). (iii) In pre-state-1, the pre-let-7/LIN28A has recruited the zinc finger of the TUT to form a ternary complex near the critical let-7 pre-miRNA 'AGGAG' binding region. This is equivalent to TUT7 ternary conformation I (Fig. 2a,b) and to the TUT4 complex (Figs. 4 and 5), a capture complex. (iv) In the active state, the 3′ end of pre-let-7 is bound into the catalytic pocket of the TUT CM, stabilizing its orientation with respect to the TUT LIM, pre-let-7 miRNA and LIN28A. This is equivalent to TUT7 ternary conformation II and is a pre-oligo-uridylation state. (v) The oligo-uridylated pre-let-7 is released as its attachment becomes destabilized.

the absence and presence of LIN28A our structures provide a mechanistic basis for understanding the role played by LIN28A in driving oligo-uridylation and downstream degradation of pre-miRNAs. They also show that LIN28A forms direct contacts with TUT4/7 as well as with its target miRNA. The conformational states captured in this work also suggest a mechanism by which, in the presence of LIN28A, oligo-uridylation may continue in a processive manner[16,17]. Thus, the flexibility of the TUT4/7 CM evident from both the TUT7 ternary complex conformation I and TUT4 complexes (in which the density of the CM is averaged out completely) (Figs. 2a,b and 4a,b and Extended Data

Figs. 4 and 7), and from the TUT7 ternary conformation II (in which, although engaged with the pre-let-7g miRNA it has a lower resolution than the rest of the complex) (Fig. 2c,d and Extended Data Fig. 4), suggests that as further resides (>3 nt) are added to the pre-miRNA 3′ end it has the flexibility to migrate away from the rest of the complex, with the pre-let-7g itself remaining firmly in the grip of the LIM plus LIN28A to allow for enzymatic processivity, and the extending poly(U) tail being bound by ZK2 (ref. 34).

The findings described in this paper provide a framework for understanding the effects of TUT4/7 + LIN28A on pre-miRNAs, and thus for the role played by TUT4/7 and LIN28A in a wide range of cancers[17,21,23–26,29–32], but they do not show how TUT4/7 target histone mRNAs[49], or a broader range of mRNAs[50] including that of interleukin-6 (IL6)[13]. Further studies will be needed to provide an equivalent mechanistic understanding for TUT4/7 targeting these alternative substrates, and a more detailed understanding of the molecular basis for the critical and varied roles played by these enzymes in cell biology[4].

## Online content

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

## Methods

### Recombinant construct preparation and protein purification

The genes of human *TUT7* and *TUT4* were synthesized by ThermoFisher Scientific. *TUT7* and *TUT4* constructs (*TUT7*: 1–1495; *TUT4* 254–1315) were subcloned (primers listed in the Supplementary Information) into the modified pHGT-bio vector with a tobacco etch virus cleavable His$_6$-GST tag at the N terminus, and a twin Strep tag at the C terminus. TUT7/4 were expressed in *E. coli* KRX stain by growing the bacteria in TB medium at 37 °C for 12 h until an optical density at 600 nm >1 followed by addition of 0.1% Rhamnose and culturing for another 36 h at 20 °C. The cells were centrifuged at 4,000*g* and suspended in lysis buffer containing 100 mM Tris-base (pH 8.0), 300 mM NaCl, 5 mM MgCl$_2$, 50 μM ZnCl$_2$ and 5% glycerol, 1 mM phenylmethyl sulfonyl fluoride, 2 mM DTT and 10 μg ml$^{-1}$ homemade benzonase. The suspended cells were sonicated in 5 s-on-4 s-off cycles for a total of 1 h on ice followed by centrifugation at 18,000*g*, 4 °C for 40 min. The supernatant containing recombinant expressed proteins were purified using the Strep-XT 4Flow beads (IBA) followed by passage through a Heparin column (GE) aiming for removal of RNA contaminants, after which the eluted fractions were pooled together and concentrated for size-exclusion chromatography (Superdex 200 Increase 10/300 (GE)) in buffer containing 20 mM HEPES (pH 7.8), 100 mM NaCl and 2 mM dithiothreitol (DTT) at 4 °C in a cold room. Protein fractions with purity better than 90% were pooled together and concentrated with a 50 kDa cutoff Centricon (Millipore) to 2 mg ml$^{-1}$ for storage at −80 °C. TUT4 wild-type and mutant proteins were purified using a very similar method.

The gene for human *LIN28A* was synthesized by Twist Bioscience. *LIN28A* was cloned into the pHGT-bio vector (primers listed in the Supplementary Information) and expressed as an N-terminal His$_6$-GST fusion protein, which was expressed in *E. coli* KRX stain in TB medium and induced by 0.1% Rhamnose (as above). LIN28A recombinant protein was purified using Ni$^{2+}$-affinity chromatography. The high levels of contaminating nucleic acids were removed by passage through a Heparin column (GE) and eluted with a linear gradient from 0.05 to 1 M NaCl. Nucleic acid-free LIN28A was concentrated and loaded onto a size-exclusion column (Superdex 200 Increase 15/300 (GE)) equilibrated in buffer containing 20 mM HEPES (pH 7.8), 100 mM NaCl and 2 mM DTT at 4 °C in a cold room. Protein fractions with purity better than 95% were pooled together and concentrated with a 10 kDa cutoff Centricon (Millipore) to 1 mg ml$^{-1}$ for storage at −80 °C.

### PEGylation modification of TUT7

To protect the TUT7 particles from sticking to the air–water interface where they were prone to denaturation, a PEGylation modification was applied for cryo-EM sample preparation. Then 100 μl of 2 mg ml$^{-1}$ TUT7 was prepared in HEPES buffer: 20 mM HEPES (pH 7.8), 100 mM NaCl and 2 mM DTT. NHS-PEG4-Azide (ThermoFisher) was added at a final concentration of 2 mM. The PEGylation reaction was performed on ice for 2 h, quenched by adding 50 mM Tris buffer (pH 8.0) followed by size-exclusion chromatography (Superdex 200 Increase 1.5/150 (GE)). Protein fractions with purity better than 95% were pooled together and concentrated with a 50 kDa cutoff Centricon (Millipore) to 1 mg ml$^{-1}$ for storage at −80 °C.

### In vitro reconstitution of TUT7 and TUT4 complexes

The pre-let-7g was synthesized by GenScript Biotech Corporation (sequence listed in the Supplementary Information). To form its hairpin structure, pre-let-7g was first heated at 95 °C for 3 min in 20 mM HEPES (pH 7.8), 100 mM NaCl, 1 mM EDTA and then slowly cooled down to room temperature for 45 min. To prepare the TUT7–RNA complex, the purified TUT7 (300 μl at 5.8 μM, around 600 μg or 3.48 nmol) was mixed with heat-treated and cooled (folded) pre-let-7 (52 μl at 50 μM) at a molar ratio of 1:1.5 and incubated further on ice for 1 h with additional presence of 0.5 mM UTPαS, 2 mM DTT and 2 mM CaCl$_2$. The complex sample was concentrated to 100 μl and subjected to size-exclusion chromatography (Superdex 200 Increase 1.5/150 (GE)) in buffer containing 20 mM HEPES (pH 7.8), 100 mM NaCl, 2 mM DTT and 2 mM CaCl$_2$ at 4 °C in a cold room. Fractions containing TUT7/ pre-let-7g were selected and concentrated to 1 mg ml$^{-1}$ (absorbance ratio $A_{260}/A_{280}$ = 1.76).

The TUT7/pre-let-7/LIN28A complex was prepared as follows: 250 μl of 5.8 μM TUT7 was mixed with 43.5 μl of 50 μM pre-let-7 and 24 μl of 120 μM LIN28A at a molar ratio of 1:1.5:2 and incubated overnight on ice with additional presence of 0.5 mM UTPαS 2 mM DTT and 2 mM CaCl$_2$. After this, the complex sample was concentrated to 100 μl and subjected to size-exclusion chromatography (Superdex 200 Increase 1.5/150 (GE)) in buffer containing 20 mM HEPES (pH 7.8), 100 mM NaCl, 2 mM DTT and 2 mM CaCl$_2$ at 4 °C in a cold room. Fractions containing TUT7/pre-let-7g/LIN28A were selected and concentrated to 1.5 mg ml$^{-1}$ ($A_{260}/A_{280}$ = 1.72). The TUT4/pre-let-7/ LIN28A sample was prepared as follows: 338 μl of 8 μM purified TUT4 was mixed with 81 μl of 50 μM pre-let-7 and 45 μl of 120 μM LIN28A at a molar ratio of 1:1.5:2 and incubated overnight on ice with additional presence of 0.5 mM UTPαS 2 mM DTT and 2 mM CaCl$_2$ followed by subjecting to size-exclusion chromatography (Superdex 200 Increase 1.5/150 (GE)) in buffer containing 20 mM HEPES (pH 7.8), 100 mM NaCl, 2 mM DTT and 2 mM CaCl$_2$ at 4 °C in a cold room. Fractions containing TUT4/LIN28A/pre-let-7 were selected and concentrated to 1 mg ml$^{-1}$ ($A_{260}/A_{280}$ = 1.79) for storage at −80 °C.

### Uridylation activity assay

The assay substrate pre-let-7g was labeled by 3′ cyanine (3′ CY) at its 5′ end during synthesis by GenScript Biotech Inc. 3′ CY labeled pre-let-7g was first denatured at 95 °C for 3 min in a solution containing 20 mM HEPES (pH 7.8), 100 mM NaCl and 1 mM EDTA, and then slowly cooled down to room temperature for 45 min. The mono-uridylation assay was performed by mixing 1 μl of 0.2 μM TUT7/4 (wild type or mutants) with 1 μl of 1 μM 3′ CY labeled pre-let-7g in 10 μl of reaction solution containing 20 mM HEPES (pH 7.8), 100 mM NaCl, 5 mM MgCl$_2$, 50 μM ZnCl$_2$, 5% glycerol, 5 mM DTT, 0.4 U of RNAase inhibitor (ThermoFisher) and 0.5 mM UTP (ThermoFisher), followed by incubating in a heating block at 37 °C for 20 min or times of 0, 5, 10, 20 min, according to the purpose of the assay. The reaction was quenched by adding 10 μl of Terminal buffer containing 0.1% SDS and 0.1 mM EDTA (pH 8.0) followed by addition of 5 μl of loading buffer (Invitrogen) and heating at 100 °C for 15 min. The products were separated by 15% (w/v) polyacrylamide gel electrophoresis under denaturing conditions and imaged using an iBright 5000 (ThermoFisher). The oligo-uridylation assay was performed according to a similar protocol, except that 1 μl of 1 μM 3′ CY labeled pre-let-7g was preincubated with 1 μl of 10 μM LIN28A and 1 μl of 0.2 μM TUT7/4 (wild type or mutants) for 20 min on ice before adding into the 10 μl reaction solution 20 mM HEPES (pH 7.8), 100 mM NaCl, 5 mM MgCl$_2$, 50 μM ZnCl$_2$, 5% glycerol, 5 mM DTT, 0.5 mM UTP and 0.4 U RNAase inhibitor (ThermoFisher) followed by separating and imaging.

### EMSA

For the electrophoretic mobility shift assay (EMSA), the 3′ CY labeled pre-let-7 was first formed into its hairpin structure by denaturation at 95 °C for 3 min in 20 mM HEPES (pH 7.8), 100 mM NaCl, 1 mM EDTA and then slowly cooled down to room temperature for 45 min. The EMSA trials were performed by mixing 1 μl of 1 μM RNA with 1 μl of 2 μM TUT7/4 in the presence or absence of 1 μl of 3 μM LIN28A in 10 μl of volume buffer containing 20 mM HEPES (pH 7.8), 100 mM NaCl, 2 mM CaCl$_2$, 50 μM ZnCl$_2$, 5% Glycerol, 5 mM DTT, 0.5 mM UTP and 0.4 U RNAase inhibitor (ThermoFisher) on ice for 30 min followed by adding 5 μl of protein native loading buffer (Novex). Samples were separated through 3–12% native acrylamide gels in 1× TBE buffer (ThermoFisher) under 100 V and 12–16 mA for 5 h at 4 °C in a cold room followed by imaging the gels via an iBright 5000 (ThermoFisher).

### Cryo-EM specimen preparation and data acquisition

The PEGylated TUT7 was diluted to 0.4 mg ml$^{-1}$, and 0.2% glycerol and 0.002% Tween-20 were added before preparing the frozen-hydrated grids. Generally, an aliquot of 3.6 µl of sample was applied to commercial grids (Au C-flat, 2/1, 200 mesh), which were preglow discharged using a plasma cleaner (Harrick Plasma system) for 30 s at medium level plasma after 1 min of evacuation. The grids were blotted for 4 s with −6 force at 4 °C and 100% humidity using a Vitrobot (FEI), followed by plunging into liquid ethane cooled by liquid nitrogen. The TUT7/pre-let-7, TUT7/pre-let-7/LIN28A and TUT4/pre-let-7/LIN28A were diluted to 0.6, 0.8 and 1 mg ml$^{-1}$, respectively. The same method was also used for the TUT7 apo sample to prepare frozen-hydrated grids.

Cryo-EM data were collected using an FEI Titan Krios operating at 300 kV with a Gatan K3 with GIF Quantum camera (at eBIC) or Falcon 4 with GIF Quantum camera (at OPIC). All data were automatically collected using EPU software with defocus ranging from −1.5 to −3 μM. Other parameters such as magnification, total dose and frames used varied between different sample collections and are provided in Table 1.

### Image processing of electron micrographs

All datasets were subject to a similar protocol for image processing via CryoSPARC[51]. Raw micrographs were imported, followed by motion correction and calculation of the contrast transfer function. The output micrographs were curated manually to remove those images with poor image quality (for example, astigmatic, moving). Then, 500 micrographs were used for automatic particle picking to generate initial 2D templates. After several rounds of 2D classification, those particles with the best 2D classes averages were selected for Topaz training[52,53] in CryoSPARC[51] using the full dataset. After diverse 2D classification with different particle picking strategies, all good particles with decent 2D class averages were merged and then the particle images of TUT7/RNA, TUT7/RNA/LIN28A and TUT4/RNA/LIN28A were rebalanced aiming to remove excess representation of certain views. The output particles were used for ab initio reconstruction, followed by hetero-refinement. Good models with better overall structure features were selected for nonuniform refinement, followed by use of DeepEMhancer[54] to improve the quality of the local density of the map. In two cases (the binary complex of TUT7+pre-miRNA, and the ternary complex of TUT7+pre-miRNA+LIN28A (conformation II)) further refinement was necessary to gain the maps presented here. For the binary complex the number of particles used for generation of an earlier map (114,985 total images in the dataset) was increased by collection of further data using a stored sample of preformed complex, to 185,567 particle images (Extended Data Fig. 2g–l). This improved the resolution from an earlier estimate of 3.76 Å to an improved 3.55 Å. For the TUT7 ternary complex conformation II, reprocessing with the benefit of additional Topax training, rebalancing of the representation of 2D averages with removal of bad particle images, allowed an increase in the dataset of qualified particles from 140,119 to 180,310 and an increase in estimated resolution from 3.63 to 3.53 Å, which also provided for better local density features (Extended Data Fig. 5).

In all cases, the resolution was determined by gold-standard Fourier shell correlation (FSC). The local resolution estimation was calculated in Chimera[55] based on the output maps from CryoSPARC[51]. 3D conformational variability analysis was carried out using 3DVA[56].

The detailed processing for every dataset is further shown systematically and schematically elsewhere in Extended Data Figs. 2, 4 and 7.

### Model building into the cryo-EM maps

The TUT7/pre-let-7/LIN28A conformation II with the highest resolution EM density map was used for initial model building. The AlphaFold TUT7 LIM region, crystal structure of TUT7 CM (Protein Data Bank (PDB) 5W0B)[34], ZK domain of LIN28A (PDB 3TS2)[35] and predicted pre-let-7g structure generated from ViennaRNA Web Services (http://rna.tbi.univie.ac.at/cgi-bin/RNAWebSuite/RNAfold.cgi) were docked

into the TUT7 ternary complex conformation II map. The generated docked model was then processed by multiple rounds of real-space refinement in Phenix[57] and manual building in Coot[58]. Unexpected density for the TUT7 ZK1 was identified, the structure of which had not been resolved in the previous crystallographic study[33]. Therefore, a model for ZK1 obtained from AlphaFold and incorporating residues 945–983 was docked into the associated region of the EM map, where it fitted well into the density. The ZK1 model was then merged into the rest of the TUT7/pre-let-7/LIN28A model before processing through multiple rounds of real-space refinement in Phenix[57] and manual rebuilding in Coot[58]. Output models were examined using either Pymol (The PyMOL Molecular Graphics System, v.2.0 Schrödinger, LLC), Chimera[55] or ChimeraX[59]. TUT7 apo, TUT7 binary complex and TUT7 ternary complex I were built based on the TUT7 ternary complex II model. Model building for the TUT4/pre-let-7g/LIN28A structure used a similar approach. Thus, the crystal structure of the TUT4 LIM (PDB 6IW6)[33] was initially docked into the EM density along with the predicted pre-let-7g structure generated from ViennaRNA Web Services (http://rna.tbi.univie.ac.at/cgi-bin/RNAWebSuite/RNAfold.cgi) before subsequent processing through multiple rounds of real-space refinement in Phenix[57] and manual building in Coot[58] using the generated model. All models were validated using Phenix[57] and are summarized in Table 1.

### Reporting summary

Further information on research design is available in the Nature Portfolio Reporting Summary linked to this article.

## Data availability

The density maps and atomic coordinates reported in this paper have been deposited in the Electron Microscopy Data Bank (EMDB) and RCSB PDB, respectively, with accession codes as follows: TUT7 apo structure EMD-16825 and PDB ID 8OEF; TUT7 bound with pre-let-7g and UTPαS EMD-17084 and PDB ID 8OPP; TUT7 bound with pre-let-7g and LIN28A conformation I EMD-17086 and PDB ID 8OPS; TUT7 bound with pre-let-7g and LIN28A conformation II EMD-17087 and PDB ID 8OPT; TUT4 bound with pre-let-7g and LIN28A (equivalent to TUT7 ternary conformation I) EMD-17164 and PDB ID 8OST. Source data are provided with this paper.

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

## Article

## Acknowledgements

We acknowledge Diamond Light Source for access and support of the cryo-EM facilities at the UK National Electron Bio-Imaging Center (eBIC, proposal nos. EM20223 and NT21004), funded by the Wellcome Trust, Medical Research Council and Biotechnology and Biological Sciences Research Council. The Division of Structural Biology is a part of the Wellcome Centre for Human Genetics, Wellcome Trust Core grant no. 090532/Z/09/Z. EM provision was provided through the OPIC EM facility, a UK Instruct-ERIC Centre, which was founded by a Wellcome JIF award (grant no. 060208/Z/00/Z) and is supported by a Wellcome equipment grant (no. 093305/Z/10/Z). Computation was performed at the Oxford Biomedical Research Computing (BMRC) facility, a joint development between the Wellcome Centre for Human Genetics (Wellcome Trust Core Award grant no. 203141/Z/16/Z) and the Big Data Institute supported by Health Data Research UK and the National Institute for Health and Care Research Oxford Biomedical Research Centre. G.Y. was a Clarendon scholar at the University of Oxford. G.Y. and R.J.C.G. were supported by the Calleva Research Centre for Evolution and Human Sciences at Magdalen College, Oxford. P.Z. was supported by the UK Wellcome Trust Investigator Award no. 206422/Z/17/Z, the UK Biotechnology and Biological Sciences Research Council grant no. BB/S003339/1 and the European Research Council AdG grant (no. 101021133). We thank Y. Zhu for discussions related to the data presented in this paper.

## Author contributions

Conceptualization was done by G.Y., C.J.N. and R.J.C.G. Methodology was developed by G.Y., M.Y., L.C., A.E.-S., T.B., P.Z. and R.J.C.G. Investigation was carried out by G.Y., M.Y., L.C. and R.J.C.G. Visualization was done by G.Y., M.Y. and R.J.C.G. Funding was acquired by T.B., P.Z. and R.J.C.G. Project administration was done by G.Y. and R.J.C.G. Supervision was performed by T.B., P.Z. and R.J.C.G. The original draft was written by G.Y. and R.J.C.G. Review and editing of the draft was carried out by G.Y., M.Y., L.C., A.E.-S., T.B., C.J.N., P.Z. and R.J.C.G.

## Competing interests

The authors declare no competing interests.

## Additional information

**Extended data** is available for this paper at https://doi.org/10.1038/s41594-024-01357-9.

**Correspondence and requests for materials** should be addressed to Robert J. C. Gilbert.

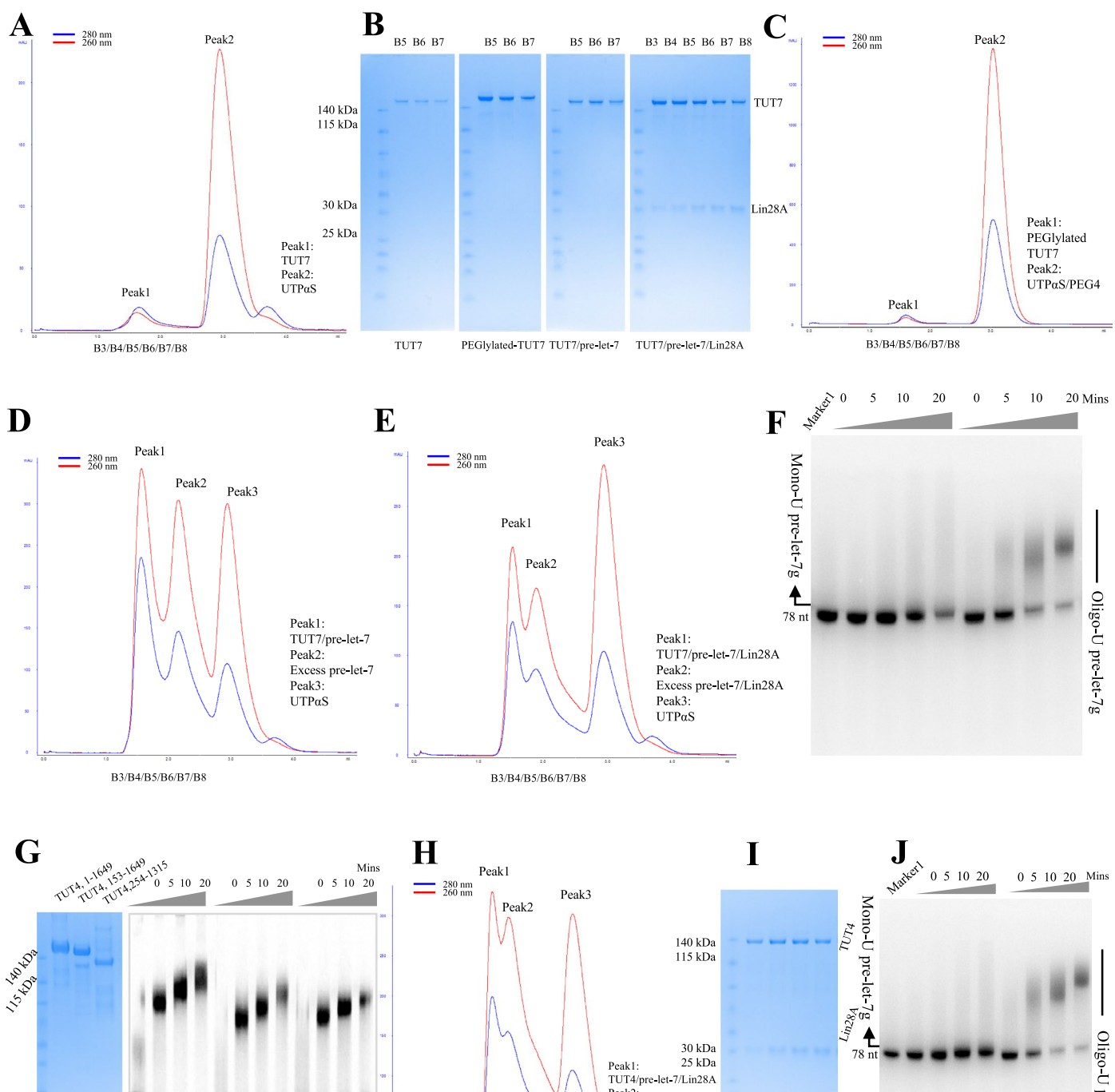

**Extended Data Fig. 1 | Assembly of TUT7 and TUT4 with pre-let-7 in the presence or absence of LIN28A.** (**a**) Size exclusion chromatography profile of apo TUT7 purification. (**b**) SDS-PAGE analysis of apo TUT7, TUT7/pre-let-7g, TUT7/pre-let-7g/LIN28A and PEGlylated TUT7 respectively. This trial performed twice. (**c**) Size exclusion chromatography of PEGlylated TUT7 in the presence of UTPαS. (**d**) Size exclusion chromatography of TUT7/pre-let-7g in the presence of UTPαS. (**e**) Size exclusion chromatography of TUT7/pre-let-7g/LIN28A. The elution volumes of peaks for apo TUT7, TUT7/pre-let-7, TUT7/pre-let-7/LIN28A and PEGlylated TUT7 are 1.62 ml, 1.54 ml, 1.52 ml and 1.7 ml respectively. (**f**) In vitro activity assays for TUT7 showing mono-uridylation activity in the absence of LIN28A or oligo-uridylation activity in the presence of LIN28A. Time across at

0, 5, 10, 20 minutes. Marker1=pre-let-7, 78 nt, shown in lane1. This array of assays performed twice. (**g**) SDS-PAGE analysis of TUT4 (254-1315), TUT4 (153-1649) and TUT4 (1-1649) (left) and in vitro activity assays for each TUT4 showing a similar level of oligo-uridylation activity in the presence of LIN28A for each expression construct. Time across at 0, 5, 10, 20 minutes (right). These assays performed twice. (**h**) Size exclusion chromatography profile of TUT4 (254-1315)/pre-let-7g/LIN28A. (**i**) SDS-PAGE analysis of TUT4(254-1315)/pre-let-7g/LIN28A. For panels H and I the assay was performed twice. (**j**) In vitro activity assays for TUT4(254-1315) showing mono-uridylation in the absence of LIN28A or oligo-uridylation activity in the presence of LIN28A. Time across at 0, 5, 10, 20 minutes, Marker1=pre-let-7, 78 nt, shown in lane1. This assay was performed twice.

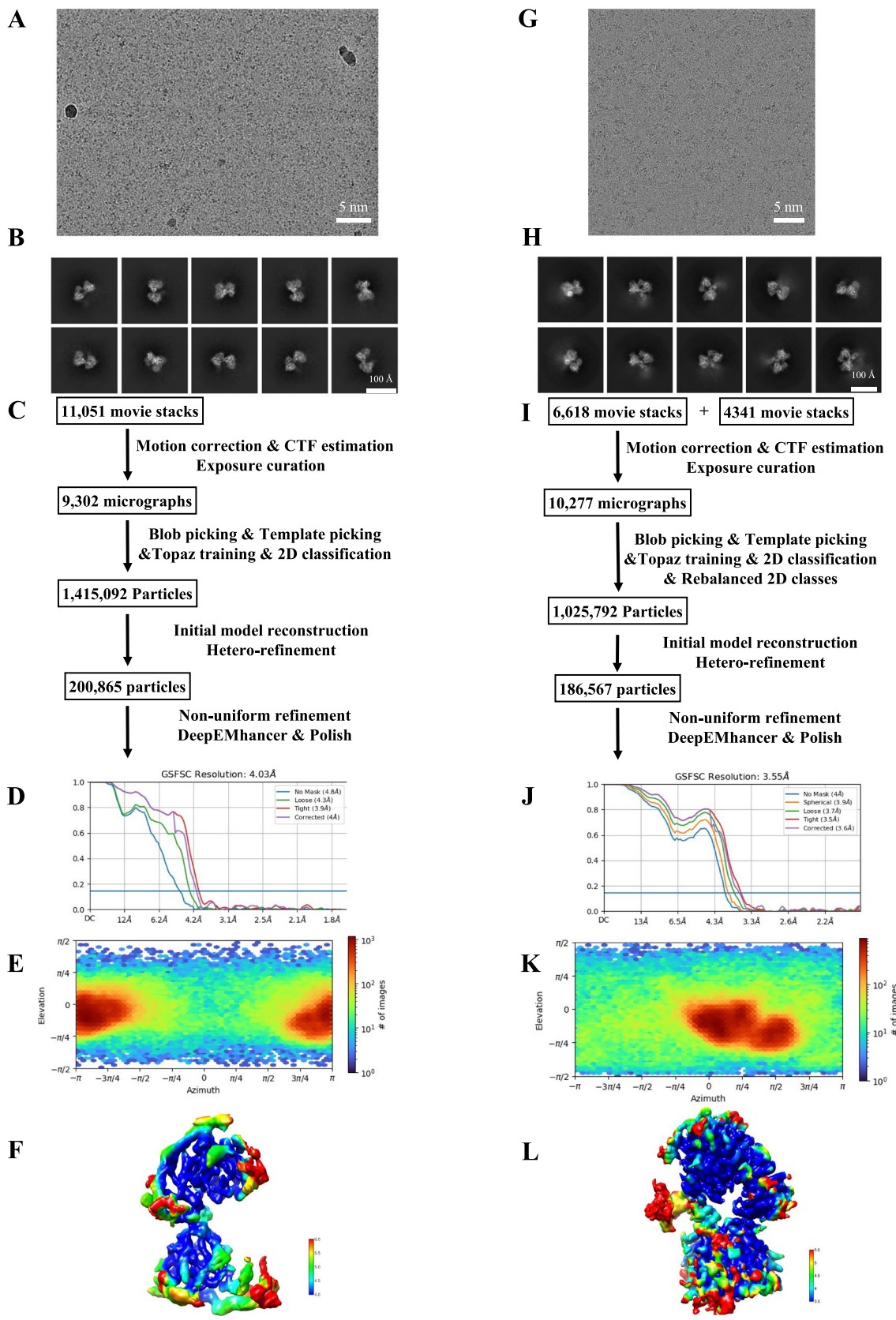

**Extended Data Fig. 2 | Cryo-EM image processing workflow of TUT7 apo and binary complex with pre-let-7 miRNA.** (**a**) Representative raw micrograph of TUT7 apo. Similar images were captured on three separate occasions. (**b**) Gallery of reference-free 2D class averages of TUT7 apo. (**c**) Data processing workflow of TUT7 apo. (**d**) Gold standard FSC curves for TUT7 apo. (**e**) Orientational distribution heat map of TUT7 apo. (**f**) Final electron density map colored according to the local resolution of TUT7 apo. (**g**) Representative raw micrograph of TUT7/pre-let-7g. Similar images were captured on two separate occasions. (**h**) Gallery of reference-free 2D class averages of TUT7/pre-let-7g. (**i**) Data processing workflow of TUT7/pre-let-7g. (**j**) Gold standard FSC curves for TUT7/pre-let-7. (**k**) Orientational distribution heat map of TUT7/pre-let-7g. (**l**) Final electron density map colored according to the local resolution of TUT7/pre-let-7g.

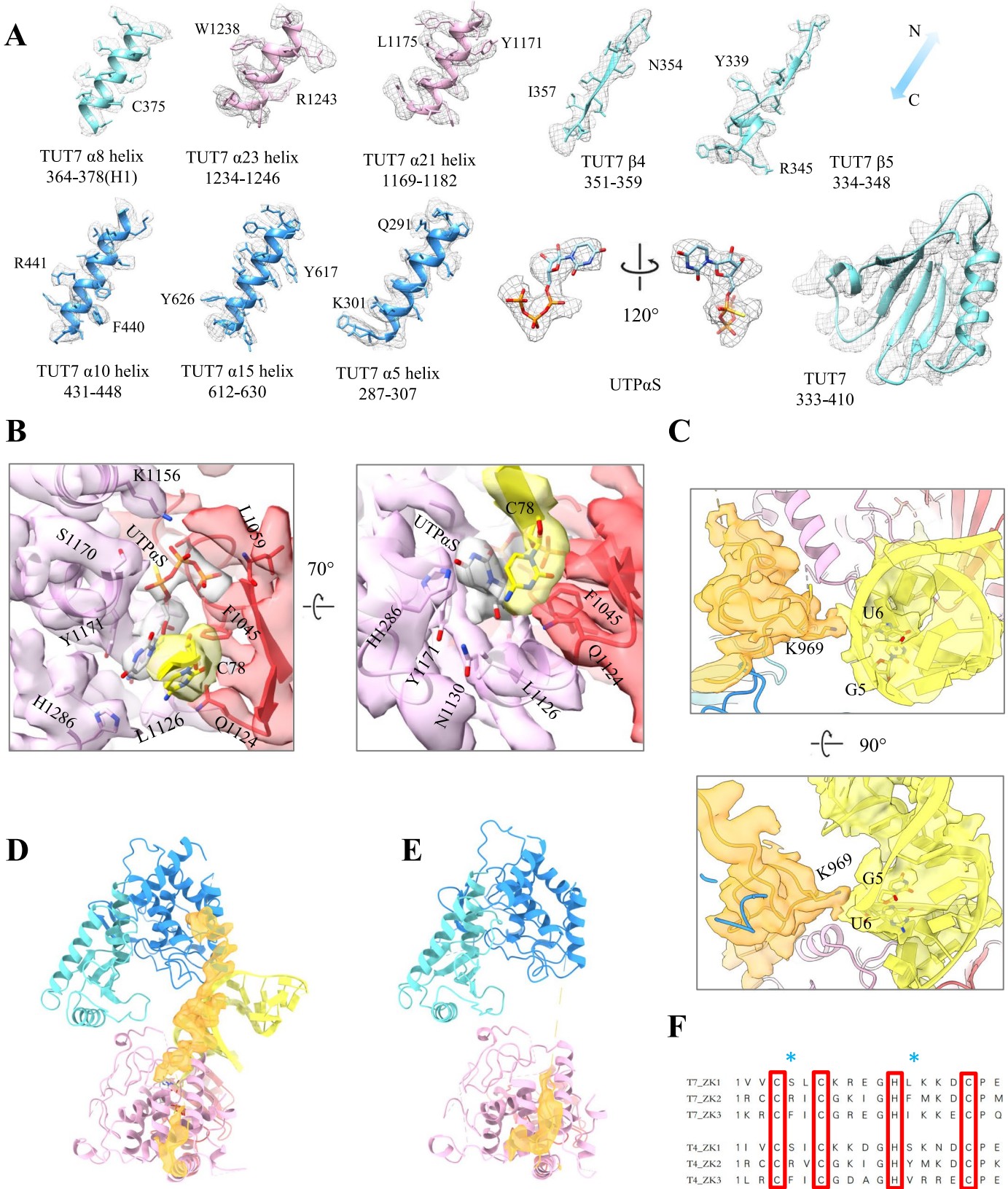

**Extended Data Fig. 3 | Structural details of the TUT7/pre-let-7g miRNA model including views of EM density maps.** (**a**) Gallery of segments from the TUT7/pre-let-7g binary complex structure showing atomic models and EM density for the protein and bound UTPαS. (**b**) Zoomed-in two views of UTPαS within the catalytic pocket interacting with the terminal pre-let-7g base C78 and the TUT7 residues shown, related by a 70° rotation, as indicated, see also Fig. 1h. (**c**) Close-up of K969 that follows the second CCHC motif cysteine in interacting with pre-let-7g in the binary complex in two views related by a 90° rotation. (**d-e**) Cartoon models of TUT7 bound with pre-let-7g miRNA (binary complex) and apo state. Coloring as in Figs. 1 and 2, with density associated with the linking polypeptide region between the LIM and CM that contains ZK1 shown in orange density map. (**f**) Sequence alignment of ZK1-ZK3 from TUT7 and TUT4; red asterisks highlight the residues after the first cysteine and first histidine in the CCHC knuckle architecture, respectively.

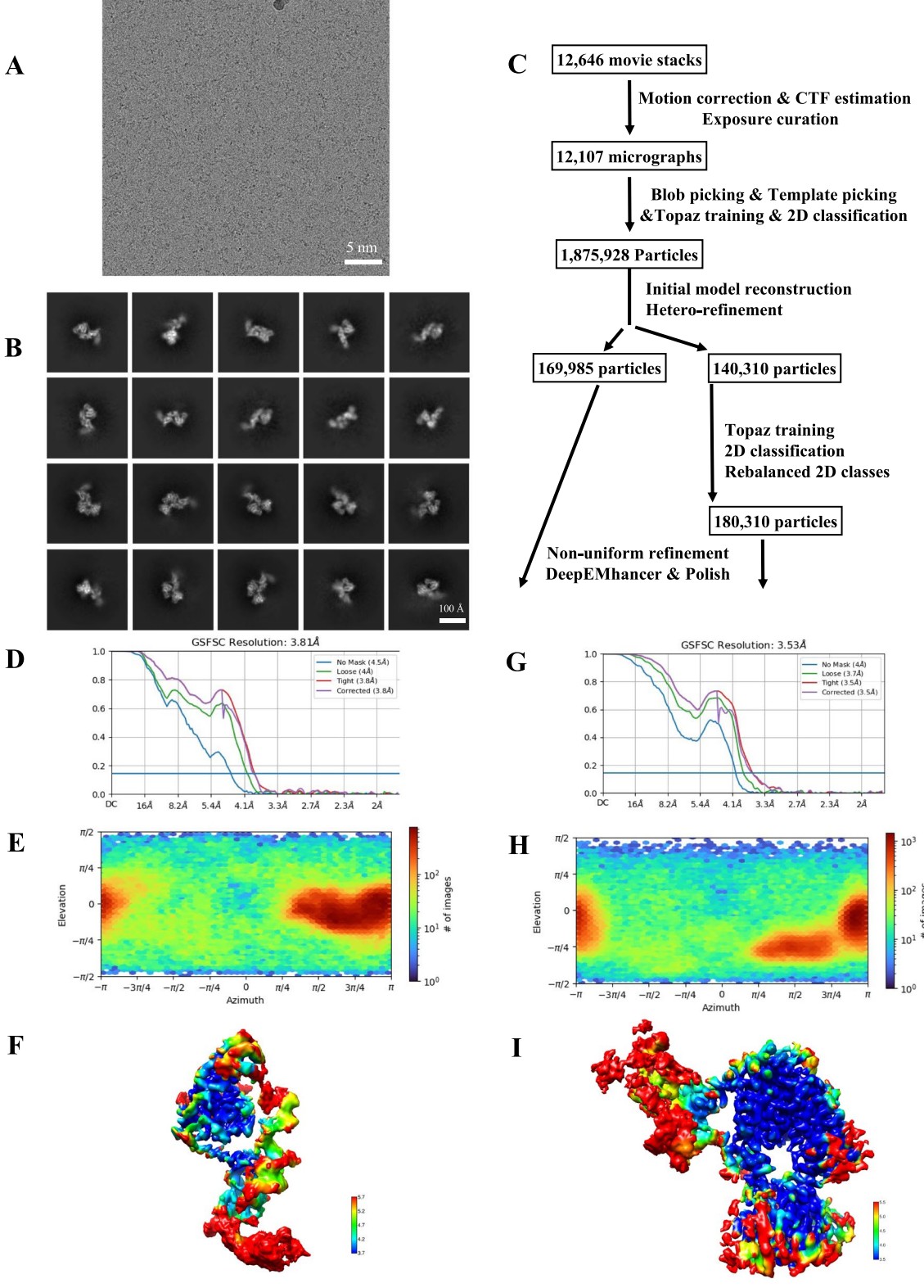

**Extended Data Fig. 4 | Cryo-EM image processing workflow of TUT7 with pre-let-7g and LIN28A.** (**a**) Representative raw micrograph of TUT7/pre-let-7g/LIN28A complexes. Similar images were captured on three separate occasions. (**b**). Gallery of reference-free 2D class averages of TUT7/pre-let-7g/LIN28A complex (**c**) Data processing workflow of TUT7/pre-let-7g/LIN28A. (**d**) Gold standard FSC curves for TUT7/pre-let-7g/LIN28A complex reconstructions (conformation-I). (**e**) Orientational distribution heat map of

TUT7/pre-let-7g/LIN28A complex (conformation-I). (**f**) Final electron density map colored according to the local resolution of TUT7/pre-let-7g/LIN28A (conformation-I). (**g**) Gold standard FSC curves for TUT7/pre-let-7g/LIN28A complex reconstructions (conformation-II). (**h**) Orientational distribution heat map of TUT7/pre-let-7g/LIN28A complex (conformation-II). (**i**) Final electron density map colored according to the local resolution of TUT7/pre-let-7g/LIN28A (conformation-II).

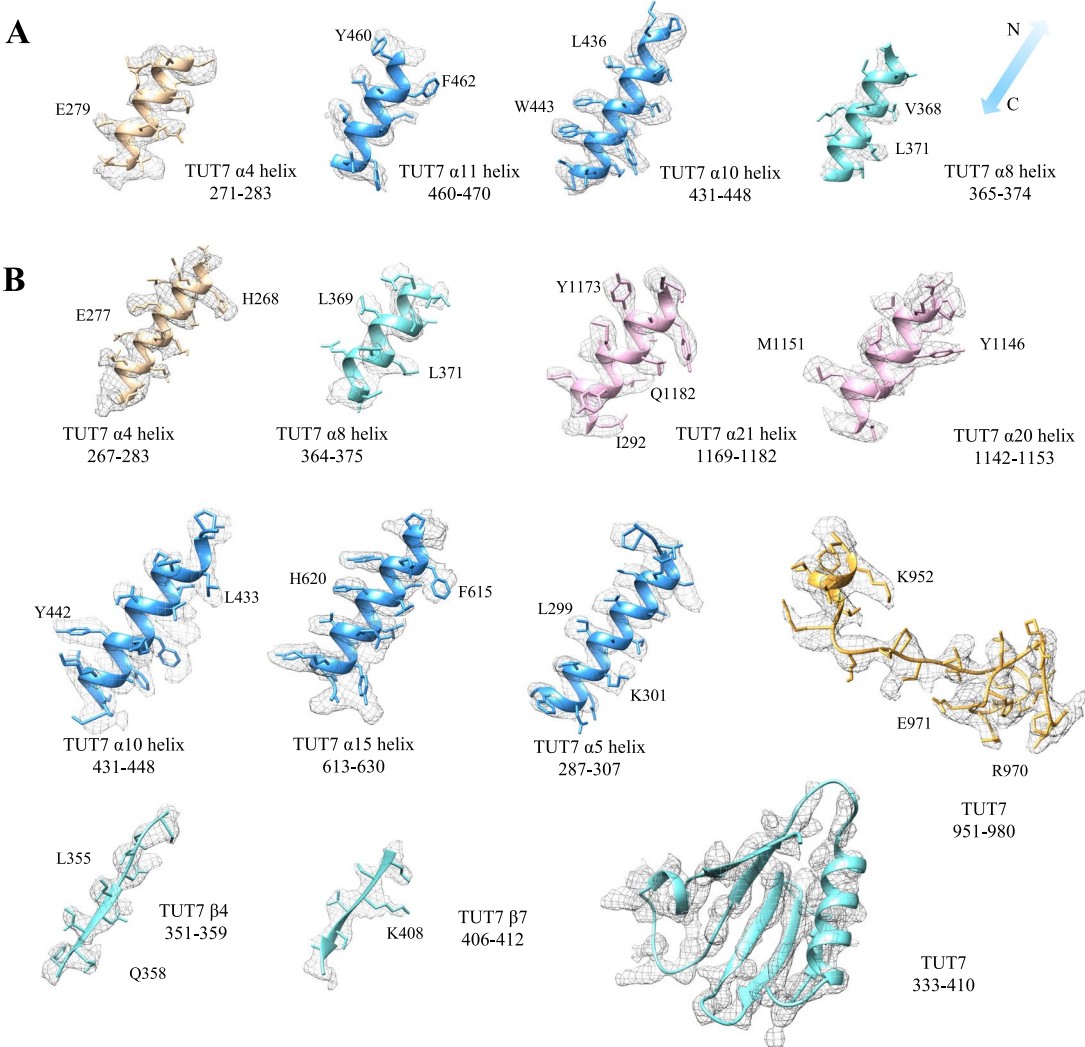

**Extended Data Fig. 5 | Representative segments of cryo-EM map fitted with the TUT7/pre-let-7g/LIN28A.** (**a**) TUT7/pre-let-7g/LIN28A conformation-I and (**b**) TUT7/pre-let-7g/LIN28A conformation-II.

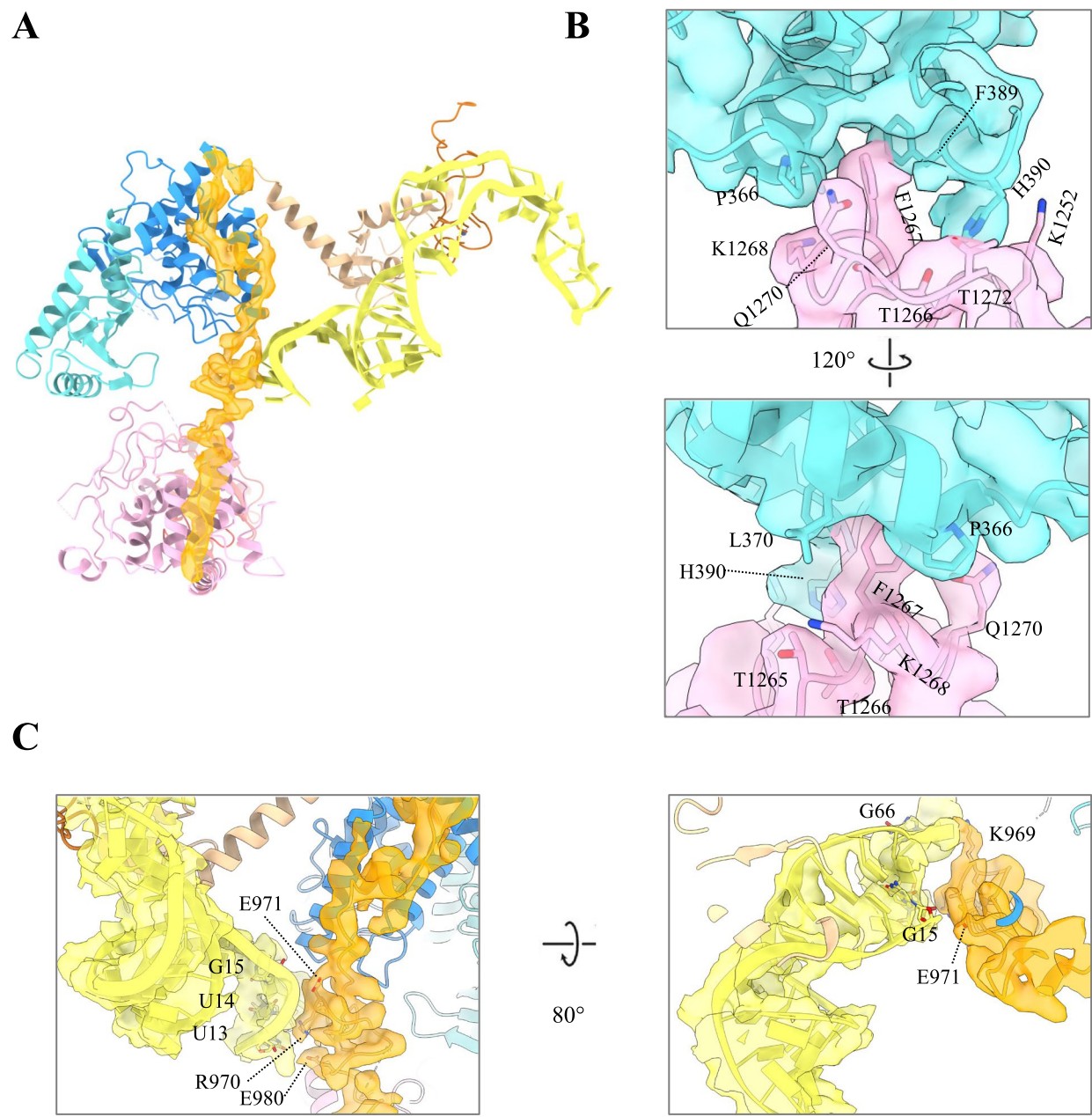

**Extended Data Fig. 6 | Structural details of TUT7/pre-let-7g/LIN28A model.** (**a**) Cartoon model of TUT7 ternary complex in conformation-II. Coloring as in Figs. 1 and 2, with density associated with the linking polypeptide region between the LIM and CM that contains ZK1 shown in orange density map. (**b**) Close-up of the physical interface between the LIM and CM in the ternary complex in two views related by a 120° rotation. (**c**) The alternative interface found in the ternary complexes in which R970, E971, K969 and E980 interface with pre-let-7g miRNA in two views related by an 80° rotation.

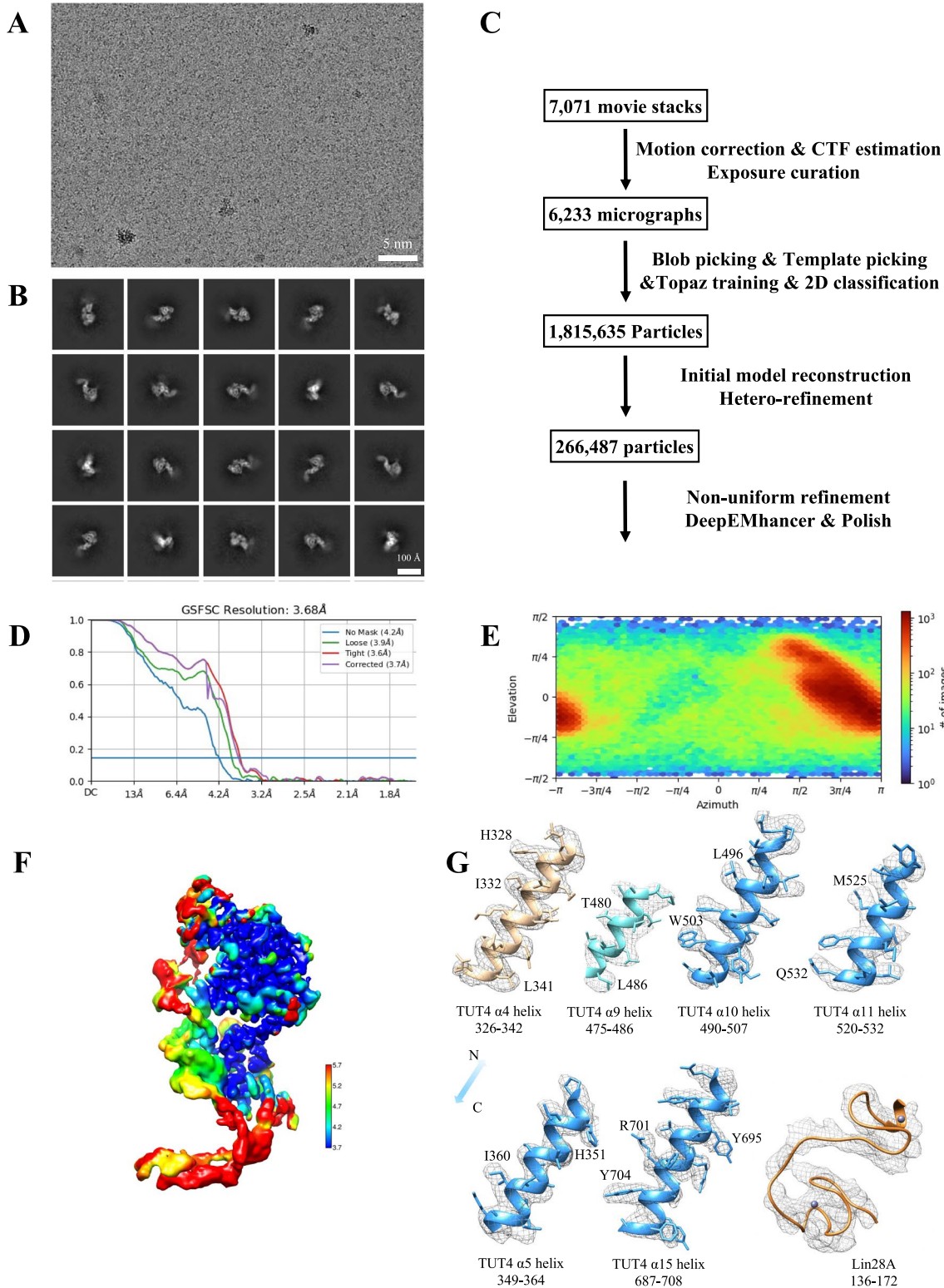

**Extended Data Fig. 7 | Cryo-EM image processing workflow of TUT4/pre-g/LIN28A.** (**a**) Representative raw micrograph of TUT4/pre-let-7g/LIN28A. Similar images were collected on four separate occasions. (**b**) Gallery of reference-free 2D class averages of TUT4/pre-let-7/LIN28A. (**c**) Data processing workflow of TUT4/pre-let-7g/LIN28A. (**d**) Gold standard FSC curves for TUT4/pre-let-7g/LIN28A. (**e**) Orientational distribution heat map of TUT4/pre-let-7g/LIN28A. (**f**) Final electron density map colored according to the local resolution of TUT4/pre-let-7g/LIN28A. (**g**) Representative segments of cryo-EM map fitted with the model of TUT4/pre-let-7g/LIN28A.

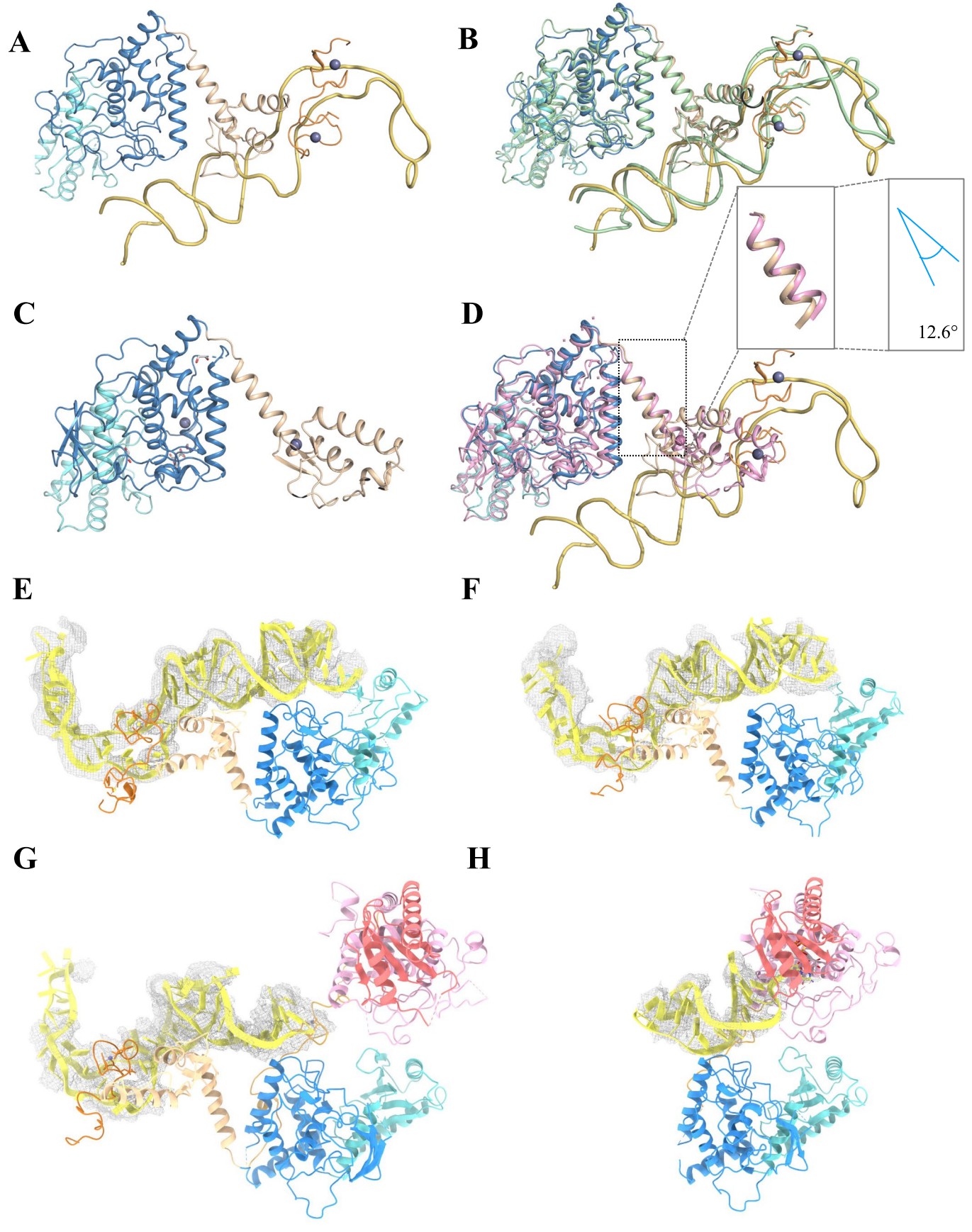

**Extended Data Fig. 8 | See next page for caption.**

**Extended Data Fig. 8 | Structural comparison of TUT4 cryo-EM and crystal structures, and of density maps for pre-let-7g in diverse states bound to TUT7 and TUT4.** (**a**) Cartoon model of the TUT4/pre-let-7g/LIN28A ternary complex reported here. (**b**) Superimposition of the TUT7/pre-let-7g/LIN28A ternary complex conformation-I (pale green) and TUT4/pre-let-7g/LIN28A ternary complex (colored), aligning by the TUT4 LIM. (**c**) Crystal structure of the TUT4 LIM (PDB: 6iw6). (**d**) Superimposition of the crystal structure of the TUT4 LIM (pink) and TUT4/pre-let-7g/LIN28A, aligning by the TUT4. (**e**) Overview of the TUT4/pre-let-7g/LIN28A complex with density for the pre-miRNA alone displayed. Protein regions are colored as in the main figures. (**f**) TUT7/pre-let-7g/LIN28A in conformation-I. (**g**) TUT7/pre-let-7/LIN28A in conformation-II. (**h**) TUT7/pre-let-7g binary complex.

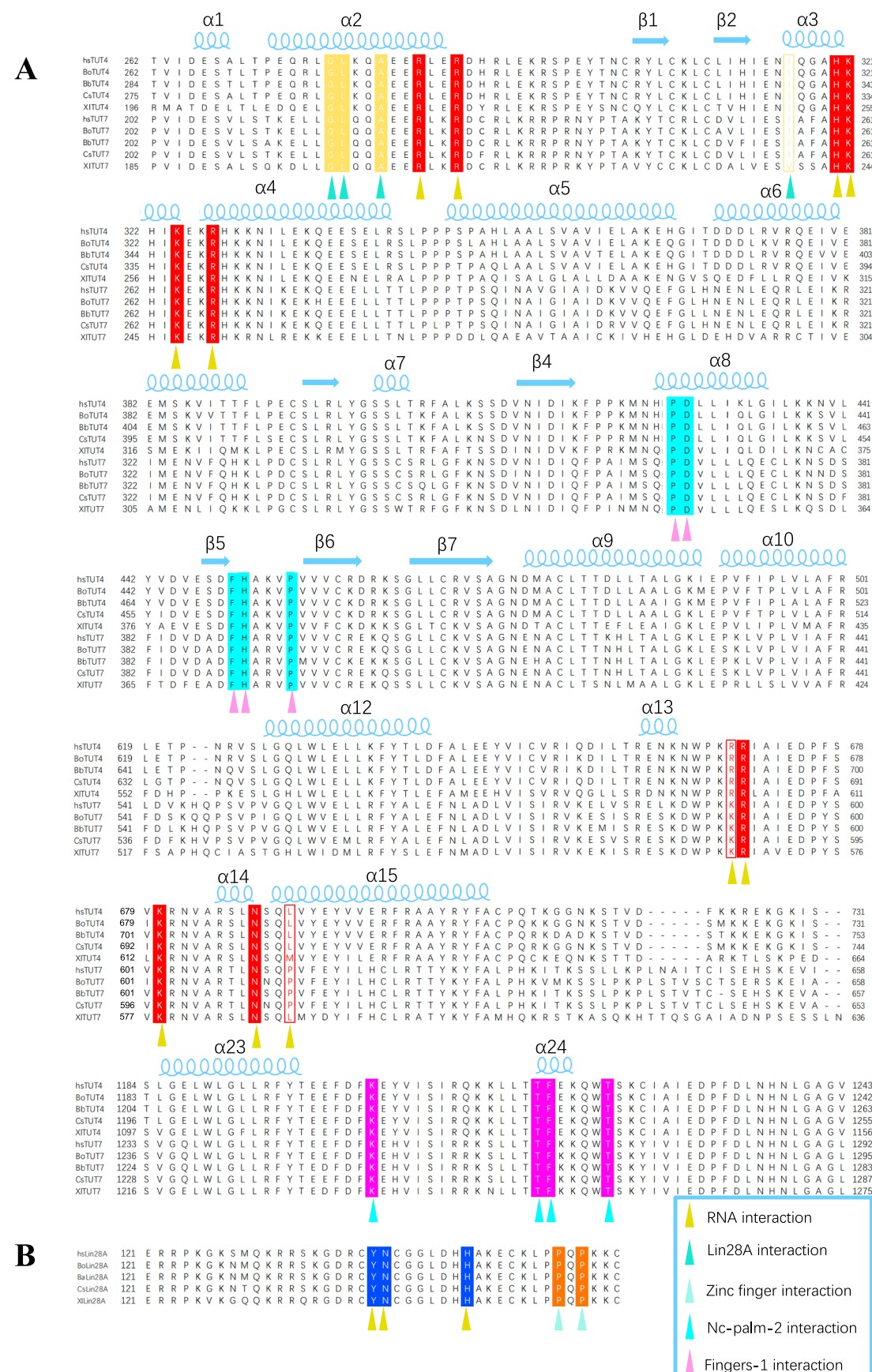

**A**

**B**

Extended Data Fig. 9 | See next page for caption.

**Extended Data Fig. 9 | Sequence alignments of TUTs and LIN28A. (a)** The amino acid sequences of human TUT4 (HsTUT4) and TUT7 (HsTUT7) are aligned with those of the TUT4 or TUT7 proteins from other organisms: California sea lion, Cow, Little brown bat and *Xenopus laevis*. Secondary structural elements revealed by TUT7 structures determined in the present study, are depicted above the sequences, with α-helices and β-sheets shown as cartoon helices and arrows, respectively. Conserved interaction site residues are shaded while non-conserved interaction site residues are colored and unshaded. **(b)** The amino acid sequence of human LIN28A (HsLIN28A) is aligned with those of the LIN28A proteins from other organisms: California sea lion, Cow, Little brown bat and *Xenopus laevis*.

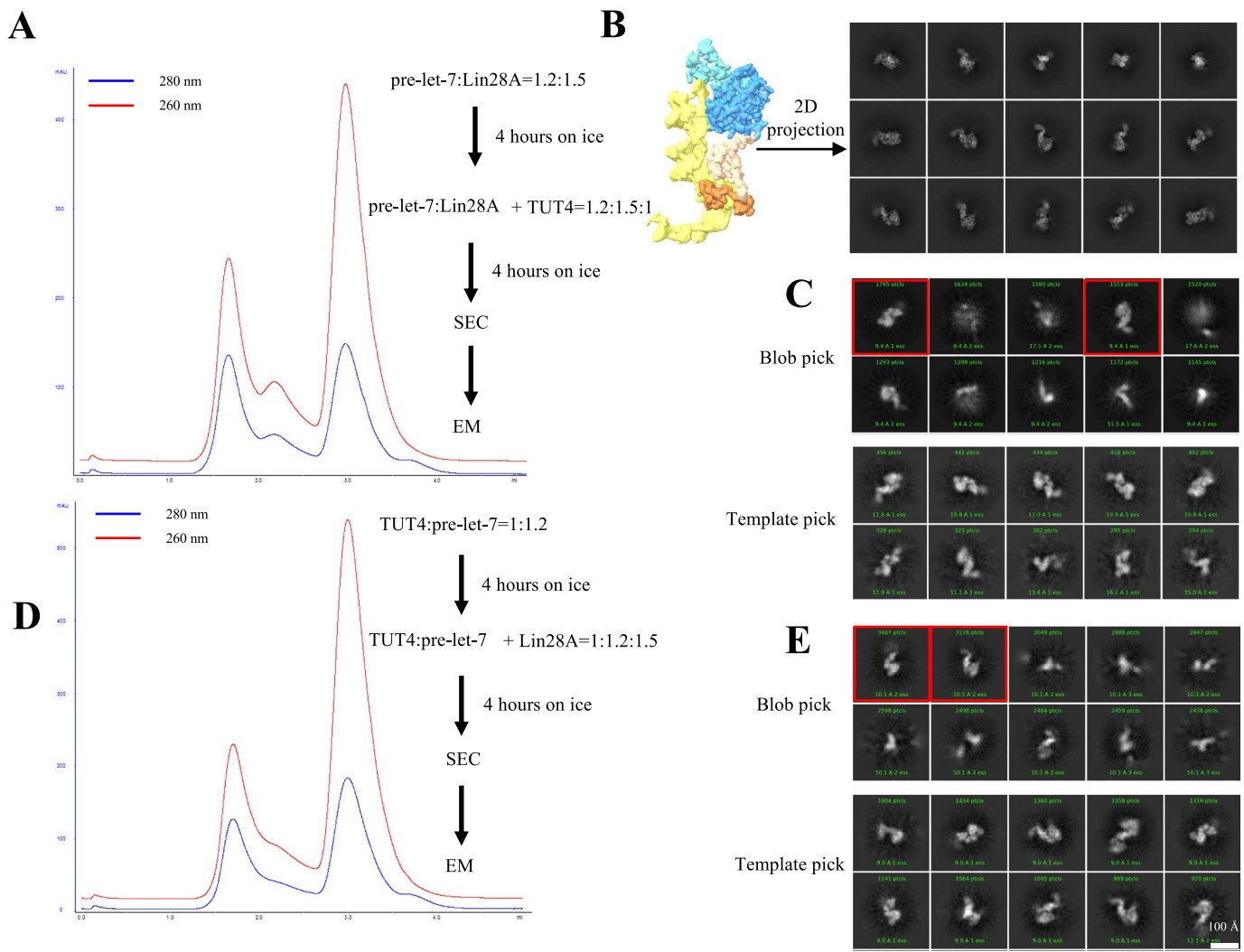

**Extended Data Fig. 10 | Assembly of TUT4 with pre-let-7 and LIN28A with components added in different orders.** (**a**) Size exclusion chromatography profile of TUT4/pre-let-7g/LIN28A assembled in order #1. The pre-let-7 and LIN28A were pre-incubated at ratio 1.2:1.5 for 4 hours on ice and then mixed with TUT4 at a final ratio 1.2:1.5:1 (pre-let-7/LIN28A/TUT4) for 4 hours on ice followed by SEC. (**b**) Representative 2D projection averages of the TUT4 3D volume described elsewhere in this paper (Figs. 4 and 5). (**c**) Gallery of reference-free 2D class averages generated with a blob picking strategy (top) and template picking strategy (bottom) for the sample assembled by order #1. Red boxed 2D class averages were chosen as templates for template picking. (**d**) Size exclusion chromatography profile of TUT4/pre-let-7g/LIN28A assembled in order #2. The TUT4 and pre-let-7 were pre-incubated at the ratio 1:1.2 for 4 hours on ice and then mixed with LIN28A at final ratio 1:1.2:1.5 (TUT4/pre-let-7/LIN28A) for 4 hours on ice followed by SEC. (**e**) Gallery of reference-free 2D class averages generated from a blob picking strategy (top) and a template picking strategy (bottom) for the sample assembled by order #2. Red boxed 2D class averages were chosen as templates for template picking.

# Reporting Summary

## Statistics

For all statistical analyses, confirm that the following items are present in the figure legend, table legend, main text, or Methods section.

| n/a | Confirmed | |
|---|---|---|
| ☐ | ☑ | The exact sample size (*n*) for each experimental group/condition, given as a discrete number and unit of measurement |
| ☒ | ☐ | A statement on whether measurements were taken from distinct samples or whether the same sample was measured repeatedly |
| ☒ | ☐ | The statistical test(s) used AND whether they are one- or two-sided<br>*Only common tests should be described solely by name; describe more complex techniques in the Methods section.* |
| ☒ | ☐ | A description of all covariates tested |
| ☒ | ☐ | A description of any assumptions or corrections, such as tests of normality and adjustment for multiple comparisons |
| ☐ | ☑ | A full description of the statistical parameters including central tendency (e.g. means) or other basic estimates (e.g. regression coefficient) AND variation (e.g. standard deviation) or associated estimates of uncertainty (e.g. confidence intervals) |
| ☒ | ☐ | For null hypothesis testing, the test statistic (e.g. *F*, *t*, *r*) with confidence intervals, effect sizes, degrees of freedom and *P* value noted<br>*Give P values as exact values whenever suitable.* |
| ☒ | ☐ | For Bayesian analysis, information on the choice of priors and Markov chain Monte Carlo settings |
| ☒ | ☐ | For hierarchical and complex designs, identification of the appropriate level for tests and full reporting of outcomes |
| ☒ | ☐ | Estimates of effect sizes (e.g. Cohen's *d*, Pearson's *r*), indicating how they were calculated |

*Our web collection on statistics for biologists contains articles on many of the points above.*

## Software and code

Policy information about availability of computer code

| Data collection | EPU Software version 2.10 was used for cryo-EM data collection. |
|---|---|
| Data analysis | CryoSPARC software version 4 was used for image analysis to determine 3D reconstructions, and DeepEMhancer version 2.0 for post-processing. Structures were built using Coot (version 0.9.8.8) and Phenix (version 1.19.2-4158), and structures were displayed using either Pymol (Version 2.0) or ChimeraX version 1.6. |

For manuscripts utilizing custom algorithms or software that are central to the research but not yet described in published literature, software must be made available to editors and reviewers. We strongly encourage code deposition in a community repository (e.g. GitHub). See the Nature Portfolio guidelines for submitting code & software for further information.

## Data

Policy information about availability of data

All manuscripts must include a data availability statement. This statement should provide the following information, where applicable:
- Accession codes, unique identifiers, or web links for publicly available datasets
- A description of any restrictions on data availability
- For clinical datasets or third party data, please ensure that the statement adheres to our policy

The density maps and atomic coordinates reported in this paper have been deposited in the EMDB and RCSB PDB, respectively, with accession codes as follows: TUT7 apo structure EMD-16825 and PDB ID 8OEF; TUT7 bound with pre-let-7g and UTPαS EMD-17084 and PDB ID 8OPP; TUT7 bound with pre-let-7g and LIN28A

conformation-I EMD-17086 and PDB ID 8OPS; TUT7 bound with pre-let-7g and LIN28A conformation-II EMD-17087 and PDB ID 8OPT; TUT4 bound with pre-let-7g and LIN28A (equivalent to TUT7 ternary conformation-I) EMD-17164 and PDB ID 8OST.

## Human research participants

Policy information about studies involving human research participants and Sex and Gender in Research.

| | |
|---|---|
| Reporting on sex and gender | N/A |
| Population characteristics | N/A |
| Recruitment | N/A |
| Ethics oversight | N/A |

Note that full information on the approval of the study protocol must also be provided in the manuscript.

# Field-specific reporting

Please select the one below that is the best fit for your research. If you are not sure, read the appropriate sections before making your selection.

☒ Life sciences     ☐ Behavioural & social sciences     ☐ Ecological, evolutionary & environmental sciences

For a reference copy of the document with all sections, see nature.com/documents/nr-reporting-summary-flat.pdf

# Life sciences study design

All studies must disclose on these points even when the disclosure is negative.

| | |
|---|---|
| Sample size | N/A |
| Data exclusions | During image analysis, image data which are of poor quality e.g. show movement or contamination, or which represent a minor conformational or compositional state are excluded, following the established CryoSPARC pipeline. |
| Replication | Assessment of resolution involves a "Gold-standard" Fourier shell correlation in which the data are separated into two halves before reconstruction, and resolution is assessed by the level of agreement between the maps. Thus replication of the structures is inherent to the analysis pipeline. |
| Randomization | Even-odd separation to two half datasets. This is standard and reckoned to be the best way to separate image data randomly. |
| Blinding | N/A |

# Reporting for specific materials, systems and methods

We require information from authors about some types of materials, experimental systems and methods used in many studies. Here, indicate whether each material, system or method listed is relevant to your study. If you are not sure if a list item applies to your research, read the appropriate section before selecting a response.

### Materials & experimental systems

| n/a | Involved in the study |
|---|---|
| ☒ ☐ | Antibodies |
| ☒ ☐ | Eukaryotic cell lines |
| ☒ ☐ | Palaeontology and archaeology |
| ☒ ☐ | Animals and other organisms |
| ☒ ☐ | Clinical data |
| ☒ ☐ | Dual use research of concern |

### Methods

| n/a | Involved in the study |
|---|---|
| ☒ ☐ | ChIP-seq |
| ☒ ☐ | Flow cytometry |
| ☒ ☐ | MRI-based neuroimaging |

