## [Peer Review File · Nature Structural & Molecular Biology]

Peer Review Information

Manuscript Title: Structural basis for activity switching in polymerases determining the fate of let-7 pre-miRNAs

Corresponding author name(s): Robert Gilbert

Reviewer Comments & Decisions:

Decision Letter, initial version:
--

Message: 29th Sep 2023

Dear Dr. Gilbert,

I am writing you on behalf of my colleague, Dr Sara Osman, who is currently on annual leave.

Thank you again for submitting your manuscript "Structural basis for activity switching in polymerases determining the fate of let-7 pre-miRNAs". I apologize for the delay in responding, which resulted from the difficulty in obtaining suitable referee reports. Nevertheless, we now have comments (below) from the 2 reviewers who evaluated your paper. In light of those reports, we remain interested in your study and would like to see your response to the comments of the referees, in the form of a revised manuscript.

You will see that all reviewers appreciate the results and find the conclusions timely and of wide interest. There are, however, several concerns and suggestions that should be addressed in a revision. Specifically, both reviewers require extensive clarification and discussion of experimental details and the proposed mechanism. Additionally, review #2 raises important technical concerns regarding the experimental protocol (pt 1) and resolution (pt 3, 7, 8). Please be sure to address/respond to all concerns of the referees in full in a point-by-point response and highlight all changes in the revised manuscript text file. If you have comments that are intended for editors only, please include those in a separate cover letter.

We expect to see your revised manuscript within 6 weeks. If you cannot send it within this

time, please contact us to discuss an extension; we would still consider your revision, provided that no similar work has been accepted for publication at NSMB or published elsewhere.

Reporting Summary:

When submitting the revised version of your manuscript, please pay close attention to our [href="https://www.nature.com/nature-portfolio/editorial-policies/image-integrity">Digital Image Integrity Guidelines](https://www.nature.com/nature-portfolio/editorial-policies/image-integrity). and to the following points below:

Please note that all key data shown in the main figures as cropped gels or blots should be presented in uncropped form, with molecular weight markers. These data can be aggregated into a single supplementary figure item. While these data can be displayed in a relatively informal style, they must refer back to the relevant figures. These data should be submitted with the final revision, as source data, prior to acceptance, but you may want to start putting it together at this point.

SOURCE DATA: we urge authors to provide, in tabular form, the data underlying the graphical representations used in figures. This is to further increase transparency in data reporting, as detailed in this editorial (<http://www.nature.com/nsmb/journal/v22/n10/full/nsmb.3110.html>). Spreadsheets can be submitted in excel format. Only one (1) file per figure is permitted; thus, for multi-paneled figures, the source data for each panel should be clearly labeled in the Excel file; alternately the data can be provided as multiple, clearly labeled sheets in an Excel file.

When submitting files, the title field should indicate which figure the source data pertains to. We encourage our authors to provide source data at the revision stage, so that they are part of the peer-review process.

Data availability: this journal strongly supports public availability of data. All data used in accepted papers should be available via a public data repository, or alternatively, as Supplementary Information. If data can only be shared on request, please explain why in your Data Availability Statement, and also in the correspondence with your editor. Please note that for some data types, deposition in a public repository is mandatory - more information on our data deposition policies and available repositories can be found below: <https://www.nature.com/nature-research/editorial-policies/reporting-standards#availability-of-data>

[Redacted]

Sincerely,

Carolina Perdigoto, PhD

Chief Editor
Nature Structural & Molecular Biology
orcid.org/0000-0002-5783-7106

Referee expertise:

Referee #1: stem cells and pluripotency

Referee #2: RNA/RNP, structural biology

Reviewers' Comments:

Reviewer #1:

Remarks to the Author:

The authors use cryoEM to illuminate the structural mechanism behind the capacity of the human TUTases (4 and 7) to mediate mono- and processive uridylation of pre-let-7 species, as dependent on the influence of LIN28. Structures of the apo states, pre-let-7 bound binary complex, and TUTase/ pre-let-7/ LIN28A ternary complexes were resolved at 3.63 to 4.03 angstroms. Full length constructs of TUT7 while truncated TUT4 were studied, but enzymatic function of both forms was documented. The binary complex reveals how TUT7 binds the pre-let-7 stem, which is inserted into the catalytic cleft with the terminal 3' cytosine overhang and bound UTPaS. The cryo structure resolves ZK1 whereas the prior crystal structure of Joshua-Tor resolved the ZK2 and suggested details about the processive oligouridylation of TUTase. The authors conclude that ZK1 stabilizes pre-let-7 for monouridylation, which by the structure shown appears plausible.

Two distinct conformations of the ternary complex were resolved. Both models resolved the interaction of the LIN28 zinc knuckle with the GGAG recognition motif in let-7 together with the zinc finger of the TUT7 LIN28 interacting domain. They performed 3D conformational variability analysis, which found stable structures for the LIN28 interacting domain of TUT7 as well as LIN28, but flexibility of the pre element of pre-let-7, consistent with adoption of different stem stabilities.

Taken together, the cryoEM structures of the binary and ternary complexes suggest that the binary complex directs the pre-let-7 stem to the catalytic module where it can undergo mono-uridylation, and is subsequently released, while in the ternary complex with LIN28 the pre-let-7 stem is directed away from the catalytic module and undergoes a two-step process of capture followed by catalytic engagement.

Higher resolution structures of the TUT4/ let-7/ LIN28 ternary complex further illustrate key elements of the interaction mechanisms, and mutagenesis of key contact residues provided support for the key role of what the authors label interface 2 and 3 for oligouridylation in the presence of LIN28.

Ultimately, the authors conclude that LIN28 interacts first with the pre-let-7, and then upon engagement with TUT7/4, provokes a large-scale conformational change in the catalytic and LIN28 interacting modules to enable the TUTase LIM zinc finger to engage

pre-let-7. From these structural insights the authors propose a model by which the binary TUT/pre-let-7 complex induces mono-uridylation whereas the ternary complex mediates oligo-uridylation. The model is plausible and adds to the structural insights from the Joshua-Tor model by resolving larger regions of TUTase interactions.

Some points to consider in the revision:

- 1) I am left wondering how these interactions can maintain oligo-uridylation of more than a few residues. How processive is the enzyme and how does uridylation extend beyond 3 residues? Some discussion is warranted.
- 2) The order of binding the authors propose in the ternary complex is Lin28/pre-let-7 first, followed by TUT7/4 binding. They need to describe more systematically their rationale for this order based on their structural studies, or provide order of addition oligouridylation studies to establish that TUT7 cannot first interact with pre-let-7 and then recruit LIN28. In LIN28 expressing cells, is LIN28 protein in excess over TUTase? If so then the order may be more driven by mass action than by structural preferences.

Minor points:

- 1) I believe the convention for human genes is to italicize in all upper case, whereas proteins are all upper case but not italicized. Authors should check and apply proper convention throughout.
- 2) For the EMSA, it is written that "3'CY labeled pre-let-7g was firstly annealed at 95 °C for 3 mins...": should annealed be changed to denatured?
- 3) To allow for replication, the authors need to provide details for the uridylation assay, as well as concentration of UTPs used for EMSA assays.
- 4) The description of the synthetic mono-U pre-let-7g vs. edited version products is vague in the method and figure 6.
- 5) "Mutagenesis of TUT4 (Figure 6A) in interfaces -2 and -3 abolished the enzyme's capacity to oligo-uridylate pre-let-7 in the presence of Lin28A"...could the authors specify which mutation was made in interface 2? (R283 – from interface 3; H320 – from interface 3; K324 – from interface 3; N688 – from interface 1); K919 –what was the logic behind introducing this mutation?

Reviewer #2:

Remarks to the Author:

Yi et al., used single-particle cryo-EM to determine the structure of the uridylyltransferases TUT4 and TUT7. The structures provide some insights into the role of the RNA-binding factor Lin28A. While the work has the potential for significant impact, the manuscript currently falls short of NSMB quality and needs to be improved before having my full support for publication.

Detailed Comments:

1- In vitro Reconstitution of TUT7 and TUT4 Complexes: The authors should provide a detailed protocol that specifies the concentrations and volumes of each component used in the in vitro reconstitution. This would enhance the reproducibility of the experiments and help readers understand the methodology better.

2- Figure Presentation: The order of figure citations is currently confusing. Figure 3A is

cited before most of the panels of Figures 1 and 2. To improve clarity, consider rearranging the figures in the manuscript to follow a more logical sequence that aligns with the flow of the text.

3- Structures and Resolution: The manuscript should explicitly state how the atomic models for the two complexes were obtained. If any predictions were used, make it clear when introducing the structure in the text. Address issues related to resolution, especially in peripheral regions, and ensure that the atomic model fits into the density. For instance, in Figure 3H part of the model needs to be better fitted into the density.

4- Binding of TUT7 to pre-let-7g miRNA (first paragraph): Figure 1H shows no residues mentioned in the text. The authors should consider improving the figures to highlight the interaction described here. Providing a close-up view that shows how the residues fit into the cryo-EM map would be helpful in supporting the findings described in the text.

5- Figure 3B is unclear. Improve its clarity and ensure that the residues fit well within the density.

6- Zinc knuckle-1 in TUT7: Considering the modest local resolution for the zinc knuckle-1 in TUT7, explain in the text how the density was assigned and the model was built.

7- Figure 3E: Based on this figure, it is unclear whether the authors have the resolution to describe the interactions mentioned in the text.

8- Enhance the representation in Figure 3E to allow readers to assess the quality of the map and determine. As presented, it is unclear whether the authors have the resolution to describe the interactions mentioned in the text.

9- Figure 3I: This is mentioned in the manuscript but not found in the figures section.

10- Superposition of TUT7 Structures: Incorporate the superposition of the two TUT7 structures as a main figure to help readers visualize the comparison.

Author Rebuttal to Initial comments

Referee expertise:

Referee #1: stem cells and pluripotency

Referee #2: RNA/RNP, structural biology

Reviewers' Comments:

Reviewer #1:

Remarks to the Author:

The authors use cryoEM to illuminate the structural mechanism behind the capacity of the human TUTases (4 and 7) to mediate mono- and processive uridylation of pre-let-7 species, as dependent on the influence of LIN28. Structures of the apo states, pre-let-7 bound binary complex, and TUTase/ pre-let-7/ LIN28A ternary complexes were resolved at 3.63 to 4.03

angstroms. Full length constructs of TUT7 while truncated TUT4 were studied, but enzymatic function of both forms was documented. The binary complex reveals how TUT7 binds the pre-let-7 stem, which is inserted into the catalytic cleft with the terminal 3' cytosine overhang and bound UTPaS. The cryo structure resolves ZK1 whereas the prior crystal structure of Joshua-Tor resolved the ZK2 and suggested details about the processive oligouridylation of TUTase. The authors conclude that ZK1 stabilizes pre-let-7 for monouridylation, which by the structure shown appears plausible.

Two distinct conformations of the ternary complex were resolved. Both models resolved the interaction of the LIN28 zinc knuckle with the GGAG recognition motif in let-7 together with the zinc finger of the TUT7 LIN28 interacting domain. They performed 3D conformational variability analysis, which found stable structures for the LIN28 interacting domain of TUT7 as well as LIN28, but flexibility of the pre element of pre-let-7, consistent with adoption of different stem stabilities.

Taken together, the cryoEM structures of the binary and ternary complexes suggest that the binary complex directs the pre-let-7 stem to the catalytic module where it can undergo monouridylation, and is subsequently released, while in the ternary complex with LIN28 the pre-let-7 stem is directed away from the catalytic module and undergoes a two-step process of capture followed by catalytic engagement.

Higher resolution structures of the TUT4/ let-7/ LIN28 ternary complex further illustrate key elements of the interaction mechanisms, and mutagenesis of key contact residues provided support for the key role of what the authors label interface 2 and 3 for oligouridylation in the presence of LIN28.

Ultimately, the authors conclude that LIN28 interacts first with the pre-let-7, and then upon engagement with TUT7/4, provokes a large-scale conformational change in the catalytic and LIN28 interacting modules to enable the TUTase LIM zinc finger to engage pre-let-7. From these structural insights the authors propose a model by which the binary TUT/pre-let-7 complex induces mono-uridylation whereas the ternary complex mediates oligo-uridylation. The model is plausible and adds to the structural insights from the Joshua-Tor model by resolving larger regions of TUTase interactions.

Some points to consider in the revision:

1) I am left wondering how these interactions can maintain oligo-uridylation of more than a few residues. How processive is the enzyme and how does uridylation extend beyond 3 residues? Some discussion is warranted.

We are grateful to the reviewer for the opportunity to clarify our thinking. We believe that the flexibility observed in the catalytic module (CM), both (i) in the existence of a conformational

state with the LIN28A-interacting module (LIM) bound to the pre-let-7g stem but in which the CM density is averaged out (TUT7 ternary complex conformation-I and TUT4 ternary complex; Figure 2A and B, Extended Data Figure 5, Figure 4A and B) [all Figure numbering is for the resubmitted version of the paper] and (ii) from the lower resolution achieved for the CM in TUT7 ternary conformation-II (Figure 2C and D, Extended Data Figure 5), suggest that the CM can adopt a variety of positions with respect to the LIM, joined by their flexible linking region (see also Extended Data Movie 3). This would allow for the conformational diversity needed for the enzyme to hold on to the pre-let-7 miRNA while the CM works at variable distances from the LIM on the 3' terminus of the pre-miRNA; work of others has highlighted a role for ZK2 in complementing the working of the CM by stabilizing an extending poly(U) tail – and this is referenced in the text. We have provided a commentary on the way in which CM flexibility might enable oligo-uridylation in the section proposing a mechanistic model (lines 348-357).

2) The order of binding the authors propose in the ternary complex is LIN28/pre-let-7 first, followed by TUT7/4 binding. They need to describe more systematically their rationale for this order based on their structural studies, or provide order of addition oligouridylation studies to establish that TUT7 cannot first interact with pre-let-7 and then recruit LIN28. In LIN28 expressing cells, is LIN28 protein in excess over TUTase? If so then the order may be more driven by mass action than by structural preferences.

We are very grateful to the reviewer on this point. We have reviewed the literature and our own data and amended the text to indicate that we cannot say what the order of binding is, indeed our data suggest any order is possible. Our original proposal was based on a number of factors which included what seemed to be the simplest interpretation, but also that LIN28A is reported to have a high affinity for pre-let-7 miRNAs – though the affinities reported in the literature vary over four orders of magnitude from 0.15 nM (ref. ¹, below), via 0.2-2 μM (refs ²⁻⁴), to 15 μM (ref. ⁵). We have also stated these measured affinities in the text (line 335). (In the papers cited affinities were measured by curves fit to data read from electromobility shift assays, which may explain the wide range of measurements made.)

As described in the Materials and Methods (lines 550-573), the approach used in the work described in the paper as originally submitted was to mix all three components together (TUT, pre-let-7 and LIN28A) at the same time, in order to form a ternary complex. We have now added new data which show that whether we add all three components at the same time or premix LIN28A + pre-let-7g before adding TUT4 (Extended Data Figure 11A), or form the binary TUT4+pre-let-7g complex and then add LIN28A (Extended Data Figure 11B), we get the same set of complexes (as assessed by size-exclusion chromatography and 2D classification of cryo-EM images).

Therefore we have modified both the text and Figure 8 in describing our proposed model for ternary complex formation of TUT+pre-let-7+LIN28A to include both modes of assembly since

both seem equally likely as far as our data show and both may occur – see lines 325-341 and Figure 8.

Minor points:

1) I believe the convention for human genes is to italicize in all upper case, whereas proteins are all upper case but not italicized. Authors should check and apply proper convention throughout. We are grateful to the reviewer for this point and have amended as indicated.

2) For the EMSA, it is written that “3’CY labeled pre-let-7g was firstly annealed at 95 °C for 3 mins...”: should annealed be changed to denatured?

We are grateful also on this point – the review is exactly right – we have changed “annealed” to “denatured” (line 577). We are no longer using the term “annealed”.

3) To allow for replication, the authors need to provide details for the uridylation assay, as well as concentration of UTPs used for EMSA assays.

We thank the reviewer for this important point. We have included a fuller set of details on sample preparation (line 550-573) and assays used (576-603).

4) The description of the synthetic mono-U pre-let-7g vs. edited version products is vague in the method and figure 6.

We have amended both the figure and the legend to make it clearer exactly which substrate/control sample is which. These remain Figure 6 and its legend in the resubmitted version.

5) “Mutagenesis of TUT4 (Figure 6A) in interfaces -2 and -3 abolished the enzyme’s capacity to oligo-uridylate pre-let-7 in the presence of LIN28A”...could the authors specify which mutation was made in interface 2? (R283 – from interface 3; H320 – from interface 3; K324 – from interface 3; N688 – from interface 1); K919 –what was the logic behind introducing this mutation?

We thank the reviewer for raising this important point. The mutations made in interface 2 were K321 and R327 (Figure 6B) and the mutation of K919 and K920 was to disrupt ZK1 binding to the open end of the pre-miRNA as these are the equivalent residues to K969 and R970 in TUT7 (see Figure 2F). We apologize that this was not made clear in the text and have amended now to make this point in the legend to Figure 6 and in the text (line 280-281). We only tested double mutants because none of the single mutations we made had a significant impact on enzymatic activity. This has resulted in interface 2 residues only being mutated in combination with a residue from another designated interface. We have again amended the text to make this point (line 274-275).

Reviewer #2:**Remarks to the Author:**

Yi et al., used single-particle cryo-EM to determine the structure of the uridylyltransferases TUT4 and TUT7. The structures provide some insights into the role of the RNA-binding factor LIN28A. While the work has the potential for significant impact, the manuscript currently falls short of NSMB quality and needs to be improved before having my full support for publication.

Detailed Comments:

1- In vitro Reconstitution of TUT7 and TUT4 Complexes: The authors should provide a detailed protocol that specifies the concentrations and volumes of each component used in the in vitro reconstitution. This would enhance the reproducibility of the experiments and help readers understand the methodology better.

We are very grateful to the reviewer for making this point and have provided detailed descriptions both for assays performed and for the process of complex assembly for structural analysis, including concentrations and volumes used (lines 550-603). In addition we have replaced Extended Data Figure 1H with an updated and correct size exclusion trace. We apologize for not providing a more complete description before now.

2- Figure Presentation: The order of figure citations is currently confusing. Figure 3A is cited before most of the panels of Figures 1 and 2. To improve clarity, consider rearranging the figures in the manuscript to follow a more logical sequence that aligns with the flow of the text.

We are again very grateful to the reviewer for this request and have substantially reorganized our data presentation. Together with other helpful requests/suggestions for changes made by the reviewer we believe the flow of our manuscript has been significantly improved. On this point in particular, we have placed what was Figure 3 into the Extended Data (as Extended Data Figures 4 and 7), and have split Extended Data Figure 3 so that no figures are called ahead of those with earlier numbers in either the main text figures or Extended Data figures (now Extended Data Figures 3 and 6). Note that to enhance understanding of the findings in the paper we have also split Extended Data Figure 5 and placed the panels relating to TUT7 into the main text, as Figure 3, while those relating to TUT4 are now Extended Figure 9. These changes address an additional point made by the reviewer (see below).

3- Structures and Resolution: The manuscript should explicitly state how the atomic models for the two complexes were obtained. If any predictions were used, make it clear when introducing the structure in the text. Address issues related to resolution, especially in peripheral regions, and ensure that the atomic model fits into the density. For instance, in Figure 3H part of the model needs to be better fitted into the density.

We thank the reviewer for this very helpful point and have provided a more detailed description of our modelling process now, in lines 651-671. In addition we have included a significantly improved pair of maps, those of the binary TUT7+pre-miRNA complex (now 3.55 Å when previously it was 3.76 Å) and of conformation-II of the ternary TUT7+pre-miRNA+LIN28A

complex (previously 3.63 Å, now 3.53 Å with an obvious improvement in map quality). These enhancements were achieved by the incorporation of more, higher-quality particle images in each case, *via* collection of an additional dataset for the binary complex and re-analysis of the ternary complex dataset previously used (see description in text, lines 634-643). The improved maps are shown in Figure 1F, Figure 2C, and Extended Data Figures 3, 4 and 6, and those updated maps and models have uploaded to the PDB and EMDB databases. The new data have enabled a significant improvement in the agreement between the model and map, such as for example in the new Extended Data Figure 4A and B where key sidechains involved in protein-protein and protein-RNA interactions are more clearly shown.

4- Binding of TUT7 to pre-let-7g miRNA (first paragraph): Figure 1H shows no residues mentioned in the text. The authors should consider improving the figures to highlight the interaction described here. Providing a close-up view that shows how the residues fit into the cryo-EM map would be helpful in supporting the findings described in the text.

With much thanks to the reviewer and the benefit of additional data giving an improved map now at 3.55 Å overall resolution we have been able to provide a more precise and accurate description of the TUT7/pre-let-7g interaction with clear sidechain density such as for K969, in Figure 1I (was 1H) with density shown in Extended Data Figure 4.

5- Figure 3B is unclear. Improve its clarity and ensure that the residues fit well within the density.

Figure 3B has now been put in the Extended Data as Extended Data Figure 4A, and with the benefit of an improved map (now 3.55 Å when previously it was 3.76 Å) has provided a better agreement between model and map. New Figures have been prepared such as Figure 1F-I, Extended Data Figure 4A (Figure 3B in previous), and Extended Data Figure 4B. Thank you again to the reviewer for this request.

6- Zinc knuckle-1 in TUT7: Considering the modest local resolution for the zinc knuckle-1 in TUT7, explain in the text how the density was assigned and the model was built.

Thanks to the reviewer and apologies we did not make this clear before. We have expanded the description of model building to include the way in which the ZK1 was identified and modelled (see lines 657-662).

7- Figure 3E: Based on this figure, it is unclear whether the authors have the resolution to describe the interactions mentioned in the text.

Thank you again to the reviewer on this point. The greatly-improved appearance of the new map has provided us with more confidence in modelling these features of the binary structure (such as of the sidechain of K969) (see Extended Data Figure 4B).

8- Enhance the representation in Figure 3E to allow readers to assess the quality of the map and determine. As presented, it is unclear whether the authors have the resolution to describe the

interactions mentioned in the text.

Thank you and the new maps and figures we hope demonstrate a significant improvement in the data (now 3.55 Å when previously it was 3.76 Å) we are reporting in this paper and also in their presentation in the map depictions.

9- Figure 3I: This is mentioned in the manuscript but not found in the figures section.

We apologize – this was a typographical error and has been corrected.

10- Superposition of TUT7 Structures: Incorporate the superposition of the two TUT7 structures as a main figure to help readers visualize the comparison.

As part of the improvements we have made to the flow of the figure presentation we have moved the superposition from the Extended Data to the main text, as the new Figure 3 panels relating to TUT7 (those relating to TUT4 are now Extended Data Figure 9), and have we hope improved the data presentation to make the comparisons made clearer and more quantitative.

1. Desjardins, A., Yang, A., Bouvette, J., Omichinski, J.G. & Legault, P. Importance of the NCp7-like domain in the recognition of pre-let-7g by the pluripotency factor Lin28. *Nucleic Acids Res* **40**, 1767-77 (2012).
2. Lightfoot, H.L. et al. A LIN28-dependent structural change in pre-let-7g directly inhibits dicer processing. *Biochemistry* **50**, 7514-21 (2011).
3. Nam, Y., Chen, C., Gregory, R.I., Chou, J.J. & Sliz, P. Molecular basis for interaction of let-7 microRNAs with Lin28. *Cell* **147**, 1080-91 (2011).
4. Piskounova, E. et al. Determinants of microRNA processing inhibition by the developmentally regulated RNA-binding protein Lin28. *J Biol Chem* **283**, 21310-4 (2008).
5. Nowak, J.S. et al. Lin28a uses distinct mechanisms of binding to RNA and affects miRNA levels positively and negatively. *RNA* **23**, 317-332 (2017).

Decision Letter, first revision:

Message: Our ref: NSMB-A47898A

2nd Jan 2024

Dear Dr. Gilbert,

Thank you for submitting your revised manuscript "Structural basis for activity switching in polymerases determining the fate of let-7 pre-miRNAs" (NSMB-A47898A), and I apologize for the delay in responding. It has now been seen by the original referees and

their comments are below. The reviewers find that the paper has improved in revision, and therefore we'll be happy in principle to publish it in Nature Structural & Molecular Biology, pending minor revisions to satisfy the referees' final requests and to comply with our editorial and formatting guidelines.

We are now performing detailed checks on your paper and will send you a checklist detailing our editorial and formatting requirements in the next couple of weeks. Please do not upload the final materials and make any revisions until you receive this additional information from us.

Sincerely,
Sara

Sara Osman, Ph.D.
Associate Editor
Nature Structural & Molecular Biology

Reviewer #1 (Remarks to the Author):

I have read the revised paper and the response to my prior review. The authors have been quite responsive and have improved the paper to my satisfaction.

Reviewer #2 (Remarks to the Author):

The authors have addressed all the main concerns I initially raised, leading to a substantial improvement in the manuscript compared to its original submission. Upon reviewing the revised content provided by the authors, I no longer have any significant remaining concerns about this work. Thus, I endorse its publication in NSMB.

Final Decision Letter:

Message: 17th Jun 2024

Dear Dr. Gilbert,

We are now happy to accept your revised paper "Structural basis for activity switching in polymerases determining the fate of let-7 pre-miRNAs" for publication as an Article in Nature Structural & Molecular Biology.

Your paper will be published online soon after we receive proof corrections and will appear in print in the next available issue. You can find out your date of online publication by contacting the production team shortly after sending your proof corrections.

You can now use a single sign-on for all your accounts, view the status of all your manuscript submissions and reviews, access usage statistics for your published articles

and download a record of your refereeing activity for the Nature journals.

Please note that *Nature Structural & Molecular Biology* is a Transformative Journal (TJ). Authors may publish their research with us through the traditional subscription access route or make their paper immediately open access through payment of an article-processing charge (APC). Authors will not be required to make a final decision about access to their article until it has been accepted. Find out more about Transformative Journals

Authors may need to take specific actions to achieve compliance with funder and institutional open access mandates. If your research is supported by a funder that requires immediate open access (e.g. according to Plan S principles) then you should select the gold OA route, and we will direct you to the compliant route where possible. For authors selecting the subscription publication route, the journal's standard licensing terms will need to be accepted, including self-archiving policies. Those licensing terms will supersede any other terms that the author or any third party may assert apply to any version of the manuscript.

Sincerely,

Sara

Sara Osman, Ph.D.
Senior Editor
Nature Structural & Molecular Biology